# F-actin patches associated with glutamatergic synapses control positioning of dendritic lysosomes

Bas van Bommel[1,†], Anja Konietzny[1,†], Oliver Kobler[2] , Julia Bär[1] & Marina Mikhaylova[1,*]

## Abstract

Organelle positioning within neurites is required for proper neuronal function. In dendrites, with their complex cytoskeletal organization, transport of organelles is guided by local specializations of the microtubule and actin cytoskeleton, and by coordinated activity of different motor proteins. Here, we focus on the actin cytoskeleton in the dendritic shaft and describe dense structures consisting of longitudinal and branched actin filaments. These actin patches are devoid of microtubules and are frequently located at the base of spines, or form an actin mesh around excitatory shaft synapses. Using lysosomes as an example, we demonstrate that the presence of actin patches has a strong impact on dendritic organelle transport, as lysosomes frequently stall at these locations. We provide mechanistic insights on this pausing behavior, demonstrating that actin patches form a physical barrier for kinesin-driven cargo. In addition, we identify myosin Va as an active tether which mediates long-term stalling. This correlation between the presence of actin meshes and halting of organelles could be a generalized principle by which synapses control organelle trafficking.

**Keywords** F-actin; lysosome; microtubule; myosin; shaft synapse
**Subject Categories** Cell Adhesion, Polarity & Cytoskeleton; Neuroscience
**The EMBO Journal (2019) 38: e101183**

## Introduction

The ability of neurons to convey and store information is based on their intricate and complex architecture, which allows a multitude of synaptic cell-to-cell contacts. The most studied and most prevalent type of synaptic contacts is excitatory spine synapses, which are located on small dendritic protrusions called spines. The actin cytoskeleton, together with scaffolding molecules at the postsynaptic density (PSD), establishes the spine architecture and enables tuning of synaptic strength. Actin in the spine has been the focus of extensive research efforts, and various functions of filamentous actin (F-actin) in dendritic spines have been described in detail (Matus, 2000; Korobova & Svitkina, 2010; Svitkina *et al*, 2010; Kim

*et al*, 2015; Bär *et al*, 2016; Hlushchenko *et al*, 2016; Mikhaylova *et al*, 2018). Spinous F-actin is classified by filament type (branched or longitudinal) and functionally divided into a stable and dynamic pool. Actin dynamics are instrumental for activity-dependent structural plasticity of dendritic spines (Matsuzaki *et al*, 2004; Racz & Weinberg, 2013; Bosch *et al*, 2014; Kim *et al*, 2015; Mikhaylova *et al*, 2018). In addition to synapses located on spines, there is a smaller fraction of glutamatergic synapses formed directly on the dendritic shaft, but their contribution to neuronal function is investigated to a lesser extent (Bourne & Harris, 2011; Reilly *et al*, 2011).

Plasticity and stability of synaptic contacts rely on several factors, including transport and delivery of proteins and mRNA from the soma, local dendritic protein synthesis, surface diffusion of membrane proteins, recycling of synaptic proteins via the dendritic secretory trafficking system, and controlled disposal of "aged" molecules mediated by lysosomes, autophagosomes, and the proteasomal system (Hanus & Schuman, 2013; Mikhaylova *et al*, 2016; Bowen *et al*, 2017; Goo *et al*, 2017; Nirschl *et al*, 2017; Penn *et al*, 2017; Seipold *et al*, 2018). Various secretory trafficking organelles are present along dendrites and help to regulate the protein pool required for potentiation and stabilization of specific synaptic contacts (van Bommel & Mikhaylova, 2016).

Recently, we and others could demonstrate that synaptic activity restrains lateral movement of ERGIC (ER to Golgi intermediate compartment), dendritic Golgi satellites, and retromer, bringing them into close spatial proximity for confined processing of secretory cargo (Hanus *et al*, 2014; Mikhaylova *et al*, 2016). Also, trafficking of lysosomes is influenced by synaptic activation. Goo and colleagues showed that lysosomes move bidirectionally in dendrites and can be stalled at the base of individual spines in response to local synaptic stimulation (Goo *et al*, 2017). In addition, neuronal activity can trigger the $Ca^{2+}$-dependent fusion of lysosomes with the dendritic plasma membrane, resulting in the release of lysosomal proteases, including cathepsin B. This leads to the activation of matrix metalloproteinase 9, which is instrumental for the remodeling of the extracellular matrix that in turn allows spine growth and long-term potentiation (Padamsey *et al*, 2017).

How could such precise organelle targeting and localization be achieved? Organelle transport critically relies on the microtubule (MT) and F-actin cytoskeleton. MTs serve as tracks for long-range active transport driven by kinesin (plus-end-directed) and dynein

1 DFG Emmy Noether Group "Neuronal Protein Transport", Center for Molecular Neurobiology, ZMNH, University Medical Center Hamburg-Eppendorf, Hamburg, Germany
2 Combinatorial Neuroimaging Core Facility (CNI), Leibniz Institute for Neurobiology, Magdeburg, Germany
*Corresponding author. Tel: +49 40 7410 55815; E-mail: marina.mikhaylova@zmnh.uni-hamburg.de
†These authors contributed equally to this work

(minus-end-directed) motors, whereas actin-based myosin motors regulate short-range transport and organelle tethering (Konietzny *et al*, 2017). Processive myosins V and VI are implicated in the transport of transmembrane proteins, such as ion channels, receptors, and transporters, in and out of dendritic spines (Kneussel & Wagner, 2013; Esteves da Silva *et al*, 2015). Most intracellular cargoes are bound by multiple and different motors, which, in combination with the cytoskeletal architecture, define the transport characteristics of organelles (van Bergeijk *et al*, 2016). In a recent study, Janssen and colleagues showed that inducible recruitment of constitutively active kinesin-1 (KIF5B) to immobile somatic peroxisomes is sufficient to drive transport into the axon (Janssen *et al*, 2017). Additional recruitment of constitutively active myosin V to the same organelle rapidly anchored them at the actin-rich axon initial segment, illustrating how combinational motor activity can influence transport. Interestingly, myosin V-induced anchoring of peroxisomes was sometimes observed along the axon, in dendrites and in the soma, suggesting that it occurs whenever cargoes with active myosin V encounter actin-rich regions (Janssen *et al*, 2017). In addition to active anchoring mechanisms, the presence of dense cytoskeletal structures and other organelles can create physical obstacles for cargo trafficking (Katrukha *et al*, 2017). Moreover, dense patches of F-actin found at the base of some dendritic spines could form obstacles for microtubule growth, directing them into spines. Synaptic activity increases the amount of F-actin in spine-associated patches, which correlates with increased probability of microtubule entry (Schätzle *et al*, 2018).

In recent years, advanced microscopy techniques have led to the discovery of novel F-actin-based structures in neurites. Perhaps the most striking and extensively studied is the periodic actin-spectrin lattice found in axons, dendrites, and dendritic spine necks (Xu *et al*, 2013; D'Este *et al*, 2015; Bär *et al*, 2016; He *et al*, 2016). Furthermore, previously undescribed patches and bundles of F-actin have been observed along the lengths of dendrites, and were suggested to serve as outgrowth points for filopodia, although their function is so far unknown (Willig *et al*, 2014, Korobova & Svitkina, 2010).

Here, we have mapped the dendritic actin cytoskeleton of mature hippocampal neurons and found local enrichments of F-actin within the dendritic shaft and at the base of dendritic spines. These F-actin patches frequently co-localized with excitatory synapse markers and contain a mixture of linear and branched filaments. Additionally, we discovered that local actin enrichments act as a regulator of lysosome trafficking, as lysosomes frequently pause at these loci. We show that the F-actin mesh acts both as a passive, physical barrier, slowing down transport of vesicles driven by MT motors, and as an anchoring point for active stalling of vesicles via myosins. These findings indicate a critical role of F-actin within dendrites, orchestrating organelle transport and positioning near synaptic contacts.

## Results

### F-actin patches frequently co-localize with synaptic markers

The organization and functional relevance of the actin cytoskeleton in dendritic shafts of mature hippocampal neurons has not been investigated in much detail so far. To map the distribution of F-actin in dendrites, we initially performed immunostainings of DIV17 neurons with the F-actin dye phalloidin-Atto647N and the dendritic microtubule marker MAP2. Confocal imaging suggested the presence of F-actin enrichments in dendritic shafts (Fig EV1A). To characterize the nanoscale organization of dendritic F-actin, we employed high-resolution stimulated emission depletion (STED) imaging (Figs 1A and EV1A). In addition to the periodic cortical F-actin lattice in dendritic shafts and spine necks, we clearly observed local enrichments of F-actin adjacent to the plasma membrane, sometimes localized at the base of dendritic spines (Fig 1A). To verify the dendritic origin of these patches, we transfected DIV16 hippocampal neurons with a plasmid encoding an anti-actin nanobody (hereafter referred to as chromobody) fused to eGFP in combination with TagBFP as cell fill. Chromobodies and phalloidin-Atto647N labeled the same dendritic F-actin foci as evident from the confocal and STED images (Fig 1B). These F-actin patches develop with the maturation of primary cultured neurons, and were first detectable at approximately DIV6 (Fig EV1B). Their time of appearance coincided with synaptogenesis, and co-localization with the presynaptic marker bassoon indicated a spatial relationship with synapses (Fig EV1B). This close proximity to presynaptic sites extended to mature neurons (DIV 16) where the actin patches were also positive for bassoon (Figs 1C and EV1C). The presence of bassoon suggests that a proportion of the dendritic actin patches are part of synaptic contacts.

Both excitatory glutamatergic synapses and inhibitory GABAergic synapses can be found on the dendritic shaft and are contacted by bassoon-positive presynapses (Richter *et al*, 1999). To distinguish between them, we performed a complementary staining with excitatory and inhibitory postsynaptic markers (homer1 and gephyrin, respectively/Fig 2A–C). Analysis indicated a high degree of co-localization of phalloidin-A647N-labeled actin patches with homer1 (65%), but to a much lesser degree with gephyrin (12.5%/Fig 2D). In DIV17-21 primary neurons, F-actin patches were present with an average frequency of 3.5 per 10 μm of dendritic shaft (Fig 2D). About 23% of them were located at the base of dendritic spines (Fig 2E). Detailed analysis of 2D STED data showed that the patch size varied between 0.02 and 0.5 μm$^2$. Interestingly, patches within the dendritic shaft that contained homer1 were significantly larger than those without (Fig 2F). About one half of patches located at the base of dendritic spines contained a PSD, but their size was not influenced by the presence of homer1 (Fig 2F).

Excitatory shaft synapses of principal neurons receive little attention compared with spine synapses, but they are clearly present in principal neurons of hippocampus and cortex (Trachtenberg *et al*, 2002; Bourne & Harris, 2011). 3D reconstructions of primary neurons transfected with mRuby2, the PSD marker FingRs-PSD95-GFP, and immunolabeled for bassoon indicated the presence of both spine and shaft synapses in medial and distal dendrites of DIV17 neurons (Fig EV2A). In agreement with primary cultures, labeling of mature CA1 pyramidal neurons in hippocampal slice cultures with mRuby2 and FingRs-PSD95-GFP revealed the presence of excitatory shaft synapses on the apical dendrite and its branches (Fig EV2B, Movie EV1). To address the functionality of axon terminals on these dendritic shafts, we performed a synaptotagmin-1 antibody uptake assay (Ivanova

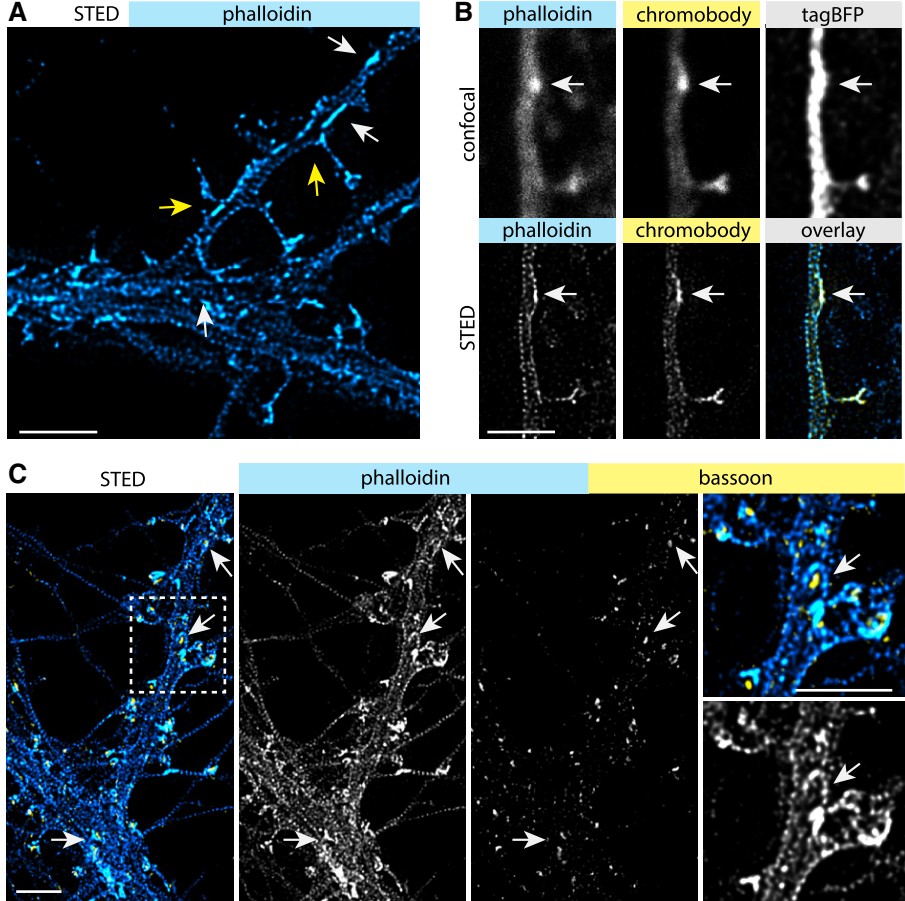

**Figure 1. Actin patches at dendritic shafts are opposed by presynaptic terminals.**

A   Deconvolved STED image of a DIV17 hippocampal primary neuron stained with phalloidin-Atto647N showing the dendritic actin cytoskeleton. The arrows indicate examples of F-actin-enriched areas, "actin patches", which can be found both at the base of dendritic spines (yellow arrows) and within dendritic shafts (white arrows). Scale bar: 2 μm.

B   Confocal and deconvolved STED image showing the co-localization of phalloidin and actin-chromobody, which label identical F-actin structures. TagBFP was expressed as cell fill. White arrow indicates F-actin patch in dendrite. Scale bar: 2 μm.

C   Deconvolved STED images showing F-actin and the presynaptic protein bassoon. The staining shows that presynaptic boutons oppose actin patches at dendritic shafts (arrows). Scale bar: 2 μm.

*et al*, 2015). DIV17 hippocampal neurons were incubated with synaptotagmin-1 antibody for 30 min, and then fixed and stained for F-actin and homer1. Confocal imaging showed that at basal synaptic activity, homer1 and F-actin-positive puncta were frequently co-localized with synaptotagmin-1, indicating that actin-enriched areas oppose active presynaptic terminals (Fig 2G). The synaptotagmin-1 signal significantly decreased when the antibody was applied to tetrodotoxin (TTX)-silenced cultures (Fig 2G and H). In addition, immunostaining of GFP-expressing neurons with antibodies against GluA1 and GluN1 indicated the presence of both α-amino-3-hydroxy-5-methyl-4-isoxazolepropionic acid (AMPA) and N-methyl-D-aspartate (NMDA) receptor subunits, indicating shaft synapses being active rather than silent synapses (Fig 2I). Taken together, these results suggest that F-actin enrichments could be a common feature for synaptic contacts on spines and dendritic shafts, and that the majority of the dendritic F-actin patches represent active excitatory shaft synapses containing both pre- and postsynaptic components.

**Dendritic F-actin patches are dynamic and consist of branched and longitudinal filaments**

In contrast to actin in spines, the structural composition of F-actin in dendritic patches is unknown (Bosch *et al*, 2014; Konietzny *et al*, 2017; Mikhaylova *et al*, 2018). In order to characterize the type of filaments present in dendritic actin patches, we performed pharmacological inhibitor experiments followed by immunostainings. Interestingly, spine-base-associated actin patches were generally more sensitive to pharmacological perturbations than shaft-associated patches: upon depolymerization of F-actin induced by a 30 min of treatment with latrunculin A (LatA; 5 μM), spine-base-associated F-actin loci disappeared almost completely, while the number and intensity of shaft-associated patches also decreased significantly, but to a much lower extent (Figs 3A and B, and EV2 C and D). We next asked what types of filaments are present at dendritic actin patches. Inhibition of the main classes of F-actin nucleators, the Arp2/3 complex (CK666; 50 μM for 2 h) and formins (SMIFH2; 30 μM for

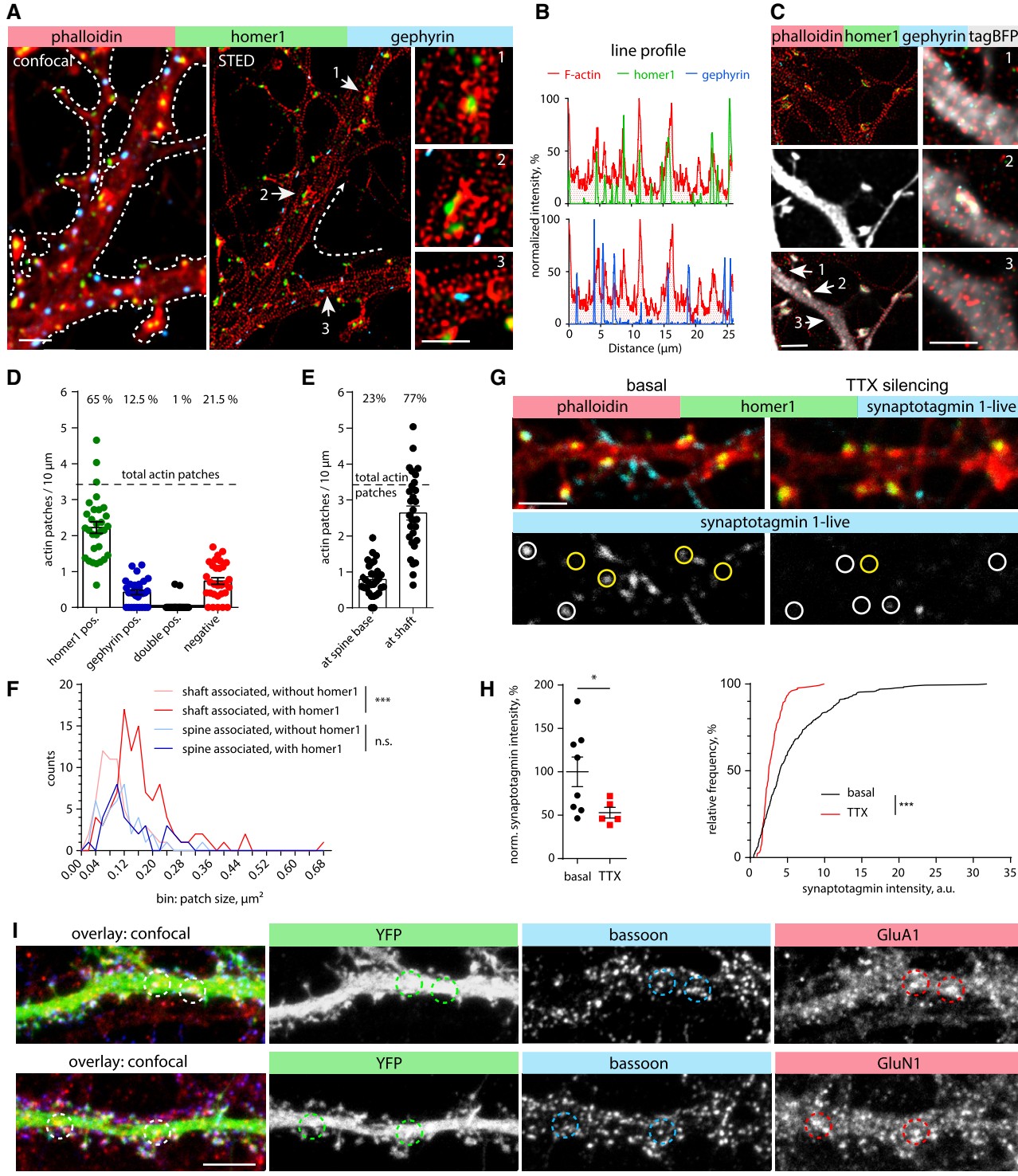

**Figure 2.**

90 min), led to a strong reduction in the number of patches localized near the base of spines and a mild decrease in shaft-associated patches (Fig 3B), while patch size and intensity remained largely unchanged (Fig EV3C and D). Moreover, both spine- and shaft-associated actin patches showed a correlation between the amount of F-actin and cortactin, a marker of branched and stable actin filaments

(Fig 3A and C). In contrast to the axonal "hotspots" described by Ganguly *et al*, which originate from F-actin nucleation by stationary endosomes, disruption of endosomes by brefeldin A treatment (100 ng/ml) had no noticeable effect on the size and localization of dendritic actin patches, suggesting an endosome-independent nature of these structures (Ganguly *et al*, 2015/Fig EV2E).

**Figure 2. The majority of actin patches are positive for excitatory synapse markers and form functional synaptic contacts.**

A   Confocal and deconvolved STED images of a DIV16 primary hippocampal neuron showing F-actin (phalloidin-Atto647N; STED), homer1 (STED), and gephyrin (confocal). Homer1 (excitatory synapse marker) and gephyrin (inhibitory synapse marker) indicate postsynaptic sites at the dendritic branch. Numbered arrows indicate regions of interest depicted with higher magnification. Zoom-in 1 and 2 show homer1 associated with dendritic actin patches. Zoom-in 3 shows gephyrin which is not associated with a dendritic actin patch. Scale bar: 2 μm, 1 μm (Zoom-in).

B   Normalized intensity profiles along the dendritic branch (line shown in panel A).

C   Confocal and deconvolved STED images of a DIV17 primary hippocampal neuron transfected with TagBFP as cell fill. Zoom-ins of the regions pointed by white arrows show gephyrin non-associated with actin patch (1), actin patches associated with homer1 (2), and single actin patches (3) located within the dendritic shaft. Scale bar: 2 μm, 1 μm (Zoom-in).

D   Description of co-localization shown in (A-C). The majority of actin patches co-localize with the excitatory synapse marker homer1 (65%), a few overlap with gephyrin (12.5%), or are negative for both markers (21.5%). n = 30 dendritic segments of 19 cells in two independent cultures. Data are presented as mean ± SEM.

E   Description of actin patches density along dendritic segments with relation to their location at the spine base or within dendritic shafts. Same n as in (D). Data are presented as mean ± SEM.

F   Frequency distribution of actin patch sizes, with or without homer1, with relation to their location at the spine base or within dendritic shafts. Dendritic actin patches within the dendritic shaft are generally larger when they contain homer1. Mann–Whitney U-test. ***$P < 0.001$. $n = 59$ (shafts associated w/o homer1), $n = 101$ (shaft associated w homer1), $n = 40$ (spine associated w/o homer1), and $n = 38$ (spine associated w homer1) patches from 27 analyzed dendritic segments of 22 cells in two independent cultures.

G   Confocal images of a DIV17 primary hippocampal neuron stained for F-actin (phalloidin) and homer1 after live uptake of α-synaptotagmin 1 antibodies by active presynaptic terminals. The presynaptic release sites show co-localization with actin patches within dendritic shafts (indicated by yellow circles). Synaptic contacts on spines are indicated by white circles. Silencing of neuronal activity by TTX application reduces synaptic release and thereby synaptotagmin labeling. Scale bar: 2 μm.

H   Quantification of synaptotagmin antibody uptake shown in (G). Left: Normalized synaptotagmin intensity measured in ROIs. Mann–Whitney U-test. *$P = 0.045$. $n = 8$ (basal) and $n = 5$ (TTX) analyzed images with each 16–82 synapses per image from 1 culture. Right: Frequency distribution of synaptotagmin intensity of individual synapses. TTX treatment significantly decreases the intensity. Two-tailed Mann–Whitney U-test. ***$P < 0.001$. $n = 303$ (basal) and $n = 132$ (TTX) synapses.

I   Maximum intensity projections of confocal image stack of YFP transfected DIV17 hippocampal primary neurons stained for bassoon (presynaptic marker), and GluA1 and GluN1 subunits. GluA1 and GluN1 show the presence of AMPA and NMDA receptors (excitatory synapse marker) on the dendritic shaft. Co-localization indicates the presence of excitatory shaft synapses (circles). Scale bar: 5 μm.

The fact that small F-actin puncta remained after LatA treatment could be an indication for a stable pool of actin filaments present in dendritic actin patches. To directly investigate the rate of actin turnover, we transfected neurons with FusionRed-actin and performed fluorescence recovery after photobleaching (FRAP) measurements. The FRAP kinetics suggests that dendritic actin patches contain dynamic and stable actin pools, with similar dynamics to actin in spines (Fig 3D and E). In addition, time-lapse imaging of F-actin labeled by expression of chromobody revealed that dendritic actin patches are relatively stable, do not move, and are persistent in their location over at least 60 min (Fig EV3A and B). In summary, dendritic F-actin patches appear to be stably localized over time and similarly to spinous F-actin consist of both branched (Arp2/3-nucleated) and linear (formin-nucleated) filaments. They have a relatively high turnover rate and contain a depolymerization-resistant, stable F-actin pool.

## Dendritic F-actin patches are spatially segregated from the microtubule cytoskeleton and control positioning of dendritic lysosomes

We hypothesized that dendritic F-actin patches act as a regulator of organelle and vesicular trafficking. We therefore decided to take a detailed look at the spatial organization of F-actin and MTs in dendrites, using STED nanoscopy on DIV17 neurons stained with phalloidin-Atto647N and α-tubulin antibodies. Intriguingly, we found that MT bundles, while closely associated with the F-actin lattice close to the cell membrane, were clearly spatially separated from dendritic actin patches rather than passing through them (Fig 4A).

To address the question whether dendritic actin patches influence organelle trafficking, we chose to look at lysosomes as a representative vesicular organelle, because they show very active, processive trafficking along dendrites, and recently, more and more

studies are emerging that highlight their significance in neuronal function (Goo *et al*, 2017; Padamsey *et al*, 2017; Cheng *et al*, 2018). First, we used 3-color STED to visualize the spatial relationship between F-actin, MTs, and lysosomes, using antibodies against the lysosomal marker protein LAMP1. We found that LAMP1 puncta were distributed all over the dendrite, with some puncta fully embedded into the F-actin mesh, while others were located next to MTs or at the interface between the F-actin and MT bundles (Fig 4B). This suggests that lysosome transport and positioning rely on both the organization of the microtubule and actin cytoskeleton.

Next, we looked at the trafficking characteristics of lysosomes in live DIV17 neurons using the fluorescent dye LysoTracker or LAMP1-eGFP expression, both labeling acidified late endosomes and lysosomes. Here, we use the term lysosomes collectively for LysoTracker or LAMP1-positive organelles. As we have shown previously that LatA treatment greatly reduces the number of dendritic actin patches (Fig 3A and B), we used the same strategy to investigate the effect of actin patches on lysosome trafficking. Using kymograph analyses, we found that depolymerization of actin filaments with LatA clearly increased the mobile fraction of lysosomes as judged by various parameters: total time spent in a mobile state increased (Fig 5A and B; Appendix Fig S1; Movie EV2), summed pausing time was reduced (Fig 5C), and pausing times were generally shorter (Fig 5D). Interestingly, directional net flux analysis indicated that LatA treatment had an effect on the directionality of trafficking (Fig 5E). On the other hand, the instant velocity and instant run length did not change (Fig 5F and G). However, a shortcoming of LysoTracker is that it indiscriminately stains all cells in a culture. To exclude lysosomal contribution from unrelated cells in the kymograph analysis, we transfected hippocampal neurons with LAMP1-eGFP and repeated the same experiments at DIV16-17. Similarly, we found that LAMP1-eGFP vesicles became more mobile following actin depolymerization (Fig 5H–J), although the distribution of individual pausing times did not change (Fig 5K). The

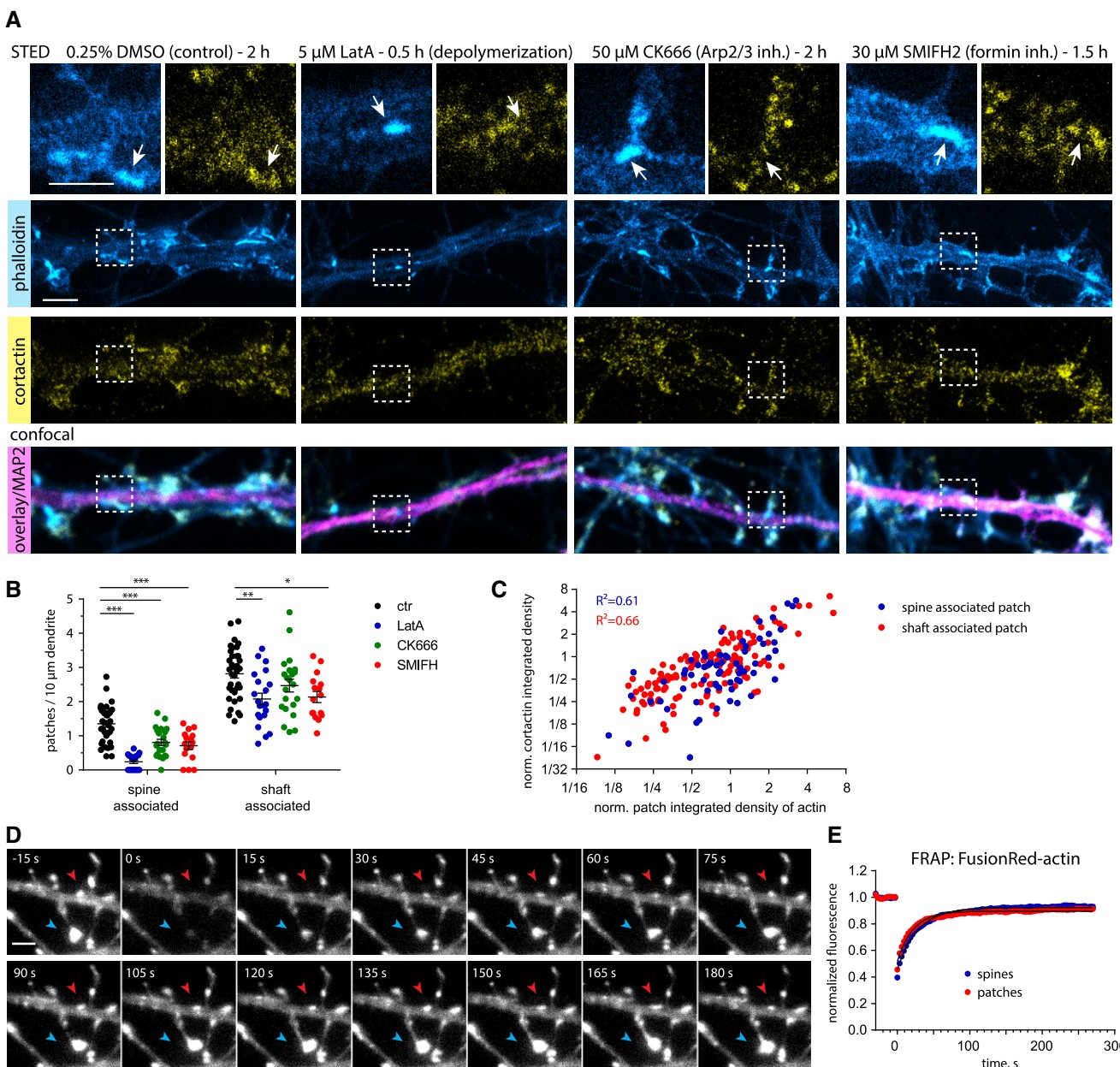

**Figure 3. Actin patches have a high actin turnover rate and contain both branched and linear filaments.**

A   Representative STED and confocal images of DIV21 rat hippocampal neurons treated with either DMSO (solvent control) for 2 h, the G-actin sequestering agent latrunculin A (LatA, 5 μM) for 0.5 h, an Arp2/3 inhibitor (CK-666, 50 μM) for 2 h, or a formin inhibitor (SMIFH2, 30 μM) for 1.5 h, and labeled for the dendritic microtubule marker MAP2, cortactin, and phalloidin-Atto647N. Zoom-ins show higher magnification image with arrow pointing to actin patches. LatA leads to the depolymerization of most, but not all, actin patches, indicating the presence of stable and non-stable actin pools within patches. Scale bar: 2 μm, 1 μm (Zoom-in).

B   Quantification of actin patches density along dendritic segments as shown in (A), with relation to their location at the spine base or within dendritic shafts. Pharmacological treatments reduced the number of spine- and shaft-associated actin patches. The reduction in both CK-666 and SMIFH2-treated samples suggest a mixed organization of branched and longitudinal actin filaments. One-way ANOVA with Dunnett's post hoc test. *$P = 0.01$, **$P = 0.002$, ***$P < 0.001$. $n = 36$ (ctr), $n = 21$ (LatA), $n = 22$ (CK666), and $n = 16$ (SMIFH2) dendritic segments of 23 (ctr), 18 (LatA), 19 (CK666), and 15 (SMIFH2) cells in three independent experiments. Data are presented as mean ± SEM.

C   Correlation of actin patches co-localizing with cortactin as shown in (A). The amount of cortactin correlates with the actin intensity for both spine- and shaft-associated actin patches (in DMSO control group). Data are plotted on a logarithmic scale. Pearson correlation. $n = 67$ (spine-associated) and $n = 132$ (shaft-associated) patches in 20 dendrites of 13 cells in two independent experiments.

D   Representative time series of a FRAP experiment with a DIV16 primary hippocampal neuron expressing FusionRed-actin. Arrows indicate fluorescence recovery photobleached FusionRed-actin in a shaft-associated actin patch (red arrow) and actin inside a dendritic spine head (blue arrow). Scale bar: 2 μm.

E   Quantification of FRAP of FusionRed-actin in actin patches and spines as shown in (D). Both actin structures display very similar recovery kinetics, indicating that shaft-associated actin patches, such as spines, contain a mobile and stable actin pool. $n = 118$ (actin patches) and $n = 129$ (spines) from 26 cells in two independent cultures.

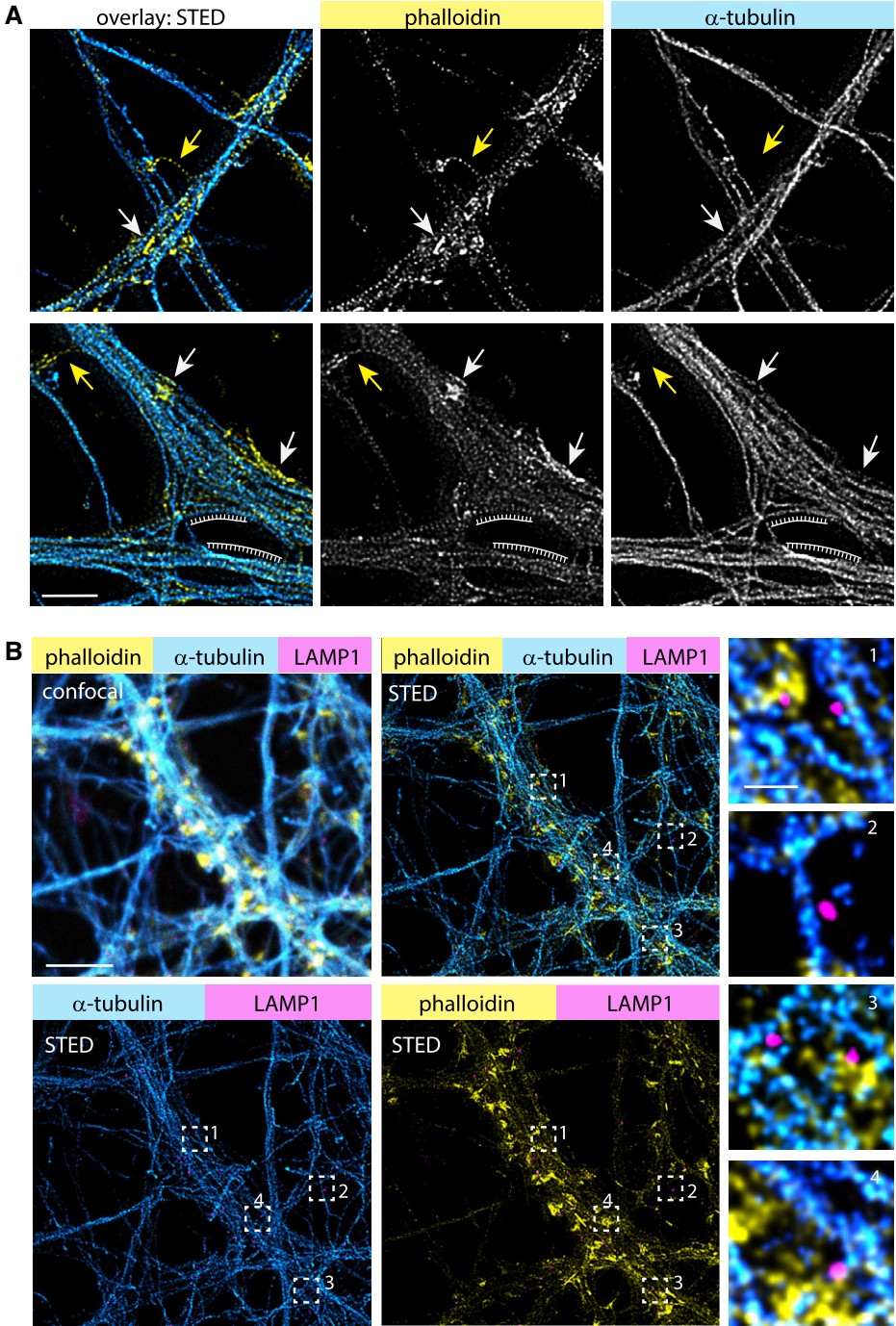

**Figure 4. Spatial organization of F-actin, microtubules, and lysosomes in dendrites.**

A   Representative deconvolved STED images of a DIV16 primary hippocampal neuron stained with phalloidin-Atto647N and α-tubulin, showing that MTs and F-actin patches are spatially segregated. Arrows indicate examples of F-actin patches in dendritic shafts (white) or F-actin rings in the spine neck (yellow). Note that MTs close to the periodic F-actin lattice in the dendrite do not show this kind of spatial separation (periodic grid). Scale bar: 2 μm.

B   Confocal and 3-color STED images of a DIV17 primary hippocampal neuron stained with phalloidin-Atto647N, α-tubulin, and α-LAMP1 antibodies. Zoom-ins shown in high magnification depict examples where lysosomes are located at the interface between F-actin and MTs (1, 4), associated with MTs (2), or embedded into the F-actin mesh (3). Scale bar: 5 μm, 0.5 μm (Zoom-in).

directional net flux was no longer biased toward anterograde trafficking (Fig 5L). Run length and instant velocity were not affected (Fig 5M and N). Overall, LAMP1-labeled vesicles were slightly more mobile than LysoTracker-stained organelles, which probably has to do with the broader spectrum of endolysosomal compartments labeled by LAMP1 (Cheng *et al*, 2018). Altogether, these data

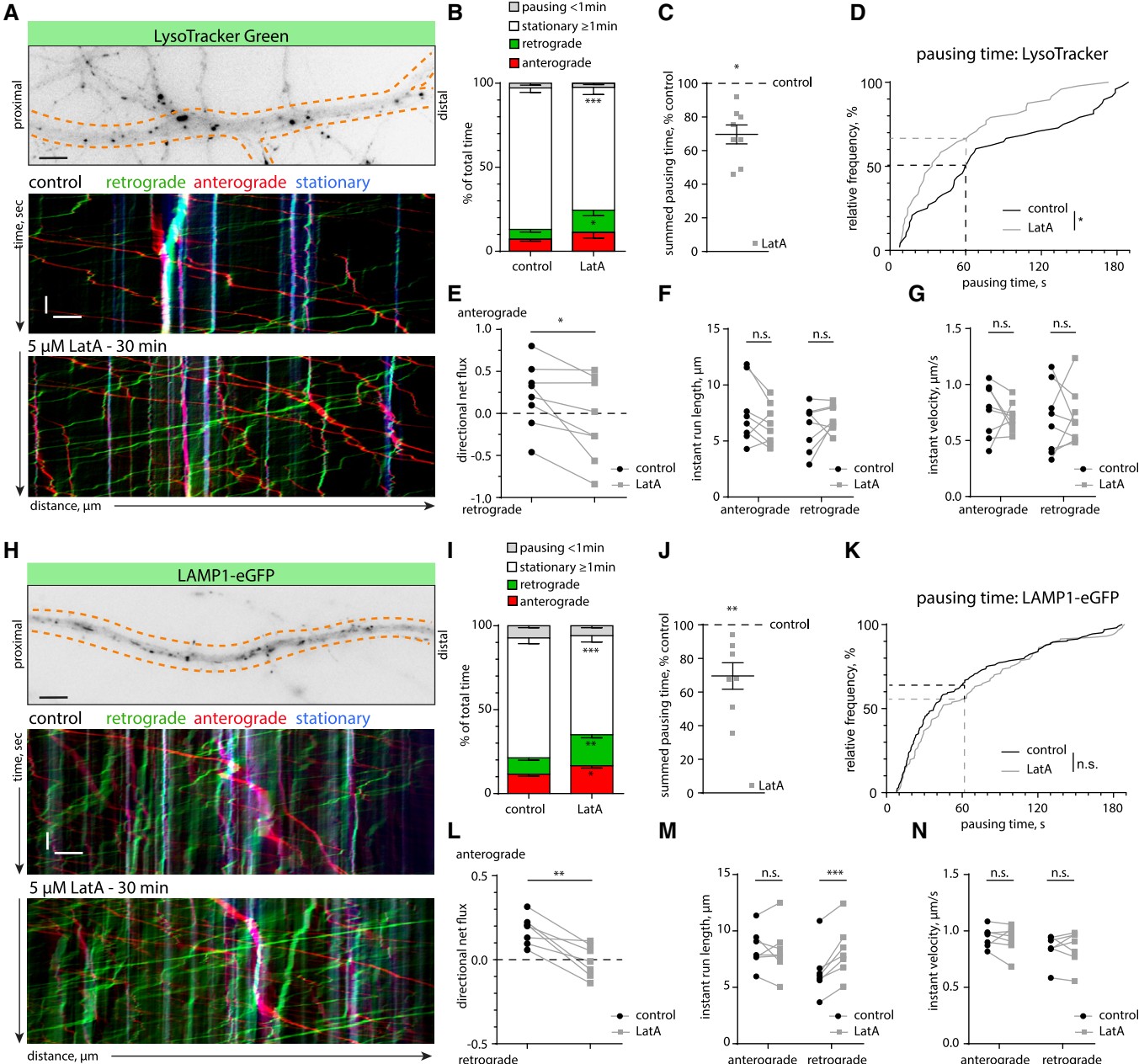

**Figure 5.**

indicate a contribution of the dendritic F-actin cytoskeleton in directing trafficking of lysosomes.

To directly observe how the presence of actin patches influences lysosomal trafficking in dendrites, we co-transfected LAMP1-eGFP and chromobody-tagRFP. Time-lapse imaging followed by kymograph analysis revealed that lysosomes frequently stop, reverse, or anchor at F-actin-rich structures located within the dendrite or at the base of dendritic spines (Fig 6A; Movie EV3). In order to obtain more quantitative information, we analyzed the average velocity of lysosomes (including stationary ones) inside or outside of actin-rich areas (Appendix Fig S2). The velocity of lysosomes in proximity to actin patches was significantly lower than in dendritic segments without accumulations of F-actin (Fig 6B).

The significant effect was lost when the mask used to mark the F-actin-enriched areas was randomly shifted (Fig 6B, Appendix Fig S2). This suggests that dense meshes of F-actin could create intracellular traps for organelles and either passively slow down MT-based trafficking or interrupt processive transport via active F-actin anchoring mechanisms.

Lysosomes are associated with different types of MT motors (kinesin-1, kinesin-2, dynein) (Pu *et al*, 2016; Farias *et al*, 2017), which makes it difficult to discriminate which motor might be primarily affected by the F-actin mesh, or if the presence of actin patches triggers competition between these motor proteins, a so-called "tug of war", resulting in stops and switches in transport directionality (Hancock, 2014). In order to be able to rule out

**Figure 5.   Latrunculin A treatment leads to an increased mobility of lysosomes.**

A  Representative image and kymographs of a time-lapse series from a dendritic segment of a DIV17 hippocampal neuron labeled with the cell-permeable lysosomal marker LysoTracker Green, before (control) and after treatment with LatA (5 μM). See also Movie EV2 and Appendix Fig S1. Scale bars: 5 μm, 10 s.

B  Quantification of lysosome motility from kymographs as shown in (A). Analyzed was the total time spent pausing (< 1 min), stationary (≥ 1 min), or moving in the anterograde or retrograde direction. LatA treatment (5 μM) increased the mobile retrograde fraction and decreased the stationary fraction. RM-2-ANOVA with behavior of lysosomes and treatment as within-group factors. $F_{(3, 21)}$ = 9.2168, $P$ = 0.0004, Newman–Keuls post hoc test: *$P$ = 0.049, ***$P$ = 0.00056. $n$ = 8 dendritic segments of six cells in three independent cultures. Data are presented as mean ± SEM.

C  Summed total pausing time of lysosomes (LysoTracker) in LatA-treated neurons, relative to the before-treatment control. LatA treatment (5 μM) decreased pausing times of lysosomes. One-sample $t$-test against 100%. *$P$ = 0.01. Same $n$ as in (B). Data are presented as mean ± SEM.

D  Cumulative frequency of the duration of pausing events. LatA treatment (5 μM) led to a shift toward shorter pausing times. Two-tailed Mann–Whitney $U$-test. *$P$ = 0.015. $n$ = 48 (pausing events) in 8 dendritic segments of six cells in three independent cultures.

E  Analysis of the directional net flux of lysosomes (LysoTracker). LatA treatment (5 μM) abolished the bias toward anterograde flux. Two-tailed paired Student's $t$-test. *$P$ = 0.02. Same $n$ as in (B).

F, G  Analysis of the instant velocity and the instant run lengths of lysosomes (LysoTracker) moving in the anterograde or retrograde direction. LatA treatment (5 μM) did not significantly affect either of these parameters. Two-tailed paired Student's $t$-test. Same $n$ as in (B).

H  Representative image and kymographs of a time-lapse series from a dendritic segment of a DIV16 hippocampal neuron transfected with LAMP1-eGFP, before (control) and after treatment with LatA (5 μM). Scale bars: 5 μm, 10 s.

I  Quantification of lysosome motility from kymographs as shown in (H). Analyzed was the total time spent pausing (< 1 min), stationary (≥ 1 min), or moving in the anterograde or retrograde direction. LatA treatment (5 μM) increased the mobile retrograde and anterograde fractions and decreased the stationary fraction. RM-2-ANOVA with behavior of lysosomes and treatment as within-group factors. $F_{(3, 18)}$ = 16.912, $P$ = 0.00002. Newman–Keuls post hoc test: *$P$ = 0.04, **$P$ = 0.005, ***$P$ = 0.0002. $n$ = 7 dendritic segments of seven cells in two independent cultures. Data are presented as mean ± SEM.

J  Summed total pausing time of lysosomes (LAMP1) in LatA-treated neurons, relative to the before-treatment control. LatA treatment (5 μM) decreased pausing time of lysosomes. One-sample $t$-test against 100%. **$P$ = 0.008. Same $n$ as in (I). Data are presented as mean ± SEM.

K  Cumulative frequency of the duration of pausing events. LatA treatment (5 μM) did not significantly change the distribution of pausing events compared with control. Two-tailed Mann–Whitney $U$-test. $n$ = 163 (ctr) and $n$ = 117 (LatA) pausing events in 7 dendritic segments of seven cells in two independent cultures.

L  Analysis of the directional net flux of lysosomes (LAMP1). Similar to the LysoTracker Green experiments (C), LatA treatment (5 μM) abolished the bias toward anterograde flux. Two-tailed paired Student's $t$-test. **$P$ = 0.003. Same $n$ as in (I).

M, N  Analysis of the instant velocity and the instant run lengths of lysosomes (LAMP1). LatA treatment (5 μM) increased the average instant run length in the retrograde direction. Two-tailed paired Student's $t$-test. ***$P$ = 0.001. Same $n$ as in (I).

tug-of-war effects, we decided to engineer a cargo that is only transported by one type of dendritic MT motor. For that, we chose the plus-end-directed motor KIF17 (kinesin-2), which is known to be involved in trafficking of dendritic secretory organelles and should therefore provide similar trafficking kinetics and pulling forces as for physiological cargo (Hanus *et al*, 2014). As a cargo, we chose peroxisomes, small and relatively stationary organelles that are mostly present in the soma (Kapitein *et al*, 2010). Peroxisomes do not associate with myosins Va and VI motors as judged by immunoblotting and immunostaining of peroxisomes enriched from rat brain (Fig EV4A and B). We artificially induced transport of peroxisomes by KIF17 via bicistronic expression of the peroxisomal targeting sequence of PEX3 fused to GFP, and a GFP nanobody (VHH_GFP) fused to constitutively active KIF17 (Fig EV4C). Live imaging of hippocampal neurons transfected with this construct revealed that a fraction of peroxisomes became mobile. The organelles were transported unidirectionally in axons and bidirectionally in dendrites, indicating that there is no additional minus-end-directed motor (i.e., dynein) attached and there will indeed be no contribution of a "tug of war" to stalling and reversals of peroxisomes (Fig EV4C). Instant velocity and instant run length were the same in anterograde and retrograde directions, confirming the model of a single type of MT motor (Fig EV4D and E). We used this construct to measure the motility of peroxisomes in the proximity of F-actin patches as we did for lysosomes. We observed that mobile peroxisomes also frequently paused at F-actin patches as evident from the kymograph (Fig 6C, Movie EV4). In contrast to lysosomes, we only analyzed peroxisomes showing at least one processive run during 3 min of imaging periods to ensure that the cargo was associated with active KIF17. However, their average speeds did not differ between inside and outside of F-actin patches (Fig 6C and D; Movie

EV3). This is probably due to the high number of short stopping events that affect the average velocity relatively little (compare Fig 5D and K control to Fig 6H control). Furthermore, we induced actin depolymerization with LatA and compared various trafficking parameters as we did for lysosomes (Fig 6E–J). LatA treatment resulted in a decrease in the summed pausing time (72% from control/Fig 6G), while the frequency distribution of the pause durations did not significantly differ from the distribution before the treatment (Fig 6H). This means that after LatA treatment, peroxisomes took fewer pauses, but the pause duration of stopped peroxisomes was unaltered. This could mean that actin patches are involved in triggering pauses, but not in long-term halting of this artificial cargo. Additionally, anterograde instant velocity and retrograde run length were reduced (Fig 6F and G). These results suggest that, similar to lysosomes, kinesin-transported peroxisomes can interact with the actin cytoskeleton. The observation that both lysosomes and KIF17-driven peroxisomes stall at actin patches excludes the possibility for a "tug of war" as a pausing mechanism. The remaining options are "passive" stalling due to physical hindrance or "active" stalling due to activation of associated actin-binding proteins such as myosin motors. As we found no evidence for neuronal peroxisomes associating with the processive myosins V and VI, and, in contrast to lysosomes, LatA treatment did not affect pause duration of peroxisomes, we decided to investigate whether these myosins might be the responsible factor to induce active stalling of lysosomes in dendrites.

**Myosin Va, but not Myosin VI, contributes to lysosome stalling**

The processive myosins V and VI are known to mediate cargo trafficking in spines. Since lysosomes can be found both in dendrites

and in dendritic spines (Goo *et al*, 2017), we asked whether lysosomes are associated with myosins and whether this contributes to stalling at actin-rich loci within dendrites. Imaging of fixed lysosomes, enriched from rat cortex and hippocampus, confirmed the presence of the lysosomal markers LAMP1 and LysoTracker (Fig 7A). Immunostaining for the processive myosins Va and VI labeled the same structures as LAMP1 (Fig 7B). Additionally, Western blotting of the lysosome-enriched fraction showed the presence of known lysosome-associated MT motor proteins (KIF5C and dynein IC1/2) (Caviston *et al*, 2011; Farias *et al*, 2017) as well as myosin Va and myosin VI (Fig 7C). These results point to the possibility that, in addition to actin patches posing as physical barriers, myosins could take over kinesin- or dynein-driven lysosomes and actively stall them once they encounter F-actin-rich environments.

To test this hypothesis, we employed a dominant negative approach to inhibit myosins V and VI. Overexpression of a dimerized tail construct of myosin Va (myoV DN), containing the cargo-binding domain, is frequently used to impair the function of endogenous myosin Va by competition (Correia *et al*, 2008; Balasanyan & Arnold, 2014; preprint: González-Gallego *et al*, 2019). Live imaging experiments of DIV16-17 neurons overexpressing mCerulean-labeled myoV DN showed visible associations between the myoV DN tail and LAMP1-positive vesicles, suggesting that lysosomes associate with myosin V motors in living cells (Fig 7D).

Kymograph analysis of LAMP1-mCherry-labeled lysosomes showed that upon overexpression of myoV DN, lysosome motility increased, as judged by the decrease in time spent being stationary (Fig 7E and G) and in pausing time distribution shifted to shorter

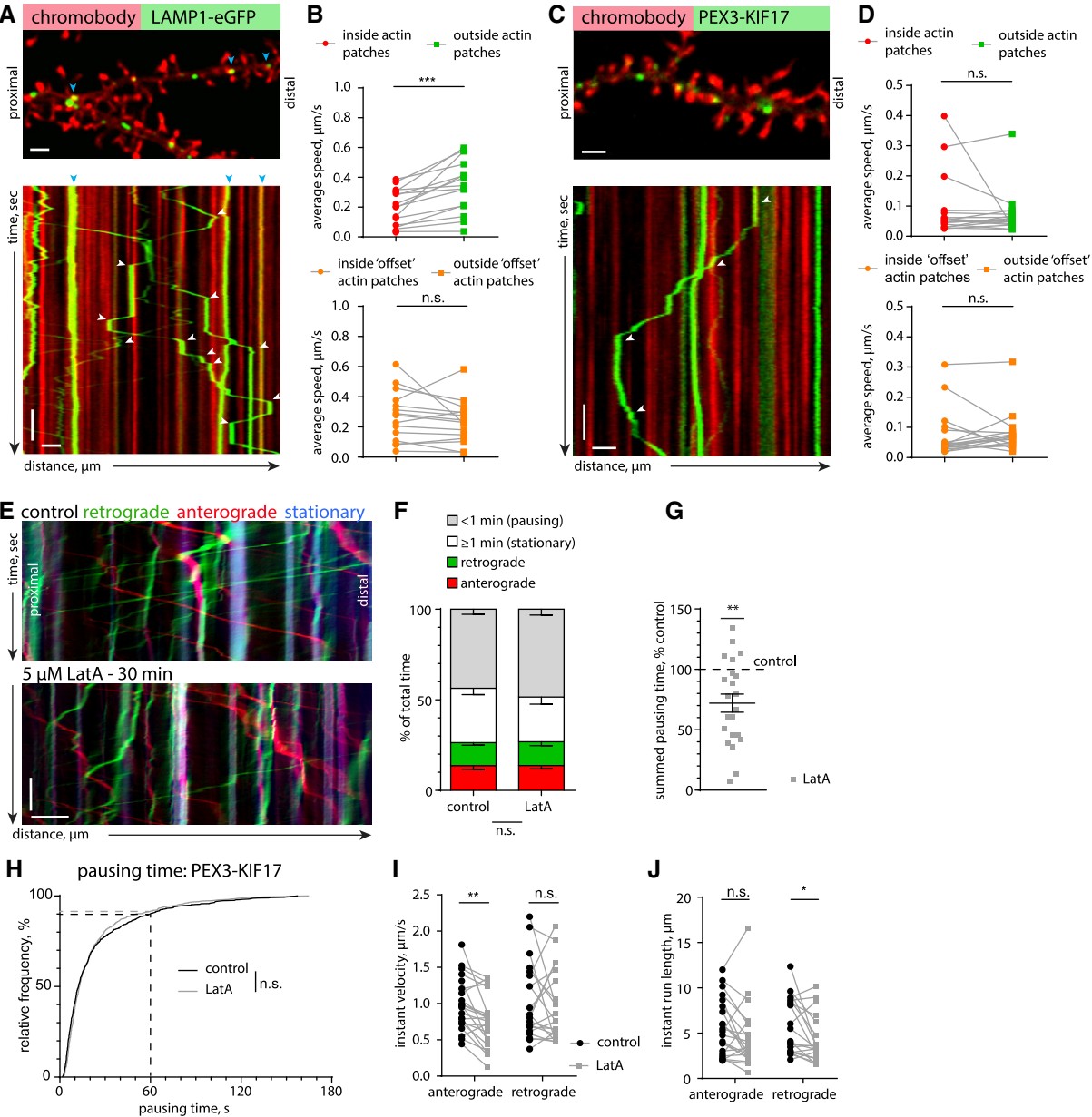

**Figure 6.**

**Figure 6. Lysosomes stall at dendritic actin patches.**

A   Representative spinning-disk confocal image and kymograph of a time-lapse series of a DIV17 hippocampal neuron transfected with the F-actin probe chromobody-tagRFP and the lysosomal marker LAMP1-eGFP. Blue arrows indicate stationary vesicles. White arrows indicate pausing events at actin patches. See also Movie EV3. Scale bar: 2 μm, 10 s.

B   Quantification of lysosome motility inside and outside actin patches. The average speed of lysosomes was reduced inside actin patches, due to pausing. As a control, randomization of the actin patch locations ("offset", as shown in Appendix Fig S2) did not give a significant result, elucidating a correlation between the location of actin patches and a reduced average speed of LAMP1-positive organelles. Two-tailed paired Student's $t$-test. ***$P$ = 0.0004. $n$ = 15 dendritic segments of 11 cells in three independent cultures.

C   Representative spinning-disk confocal image and kymograph of a time-lapse series of a DIV16 hippocampal neuron transfected with chromobody-tagRFP and PEX-GFP-KIF17 (see also Fig EV4C). The kymograph shows that KIF17-coupled peroxisomes frequently stall at actin patches. White arrows indicate stopping events at actin patches. See also Movie EV4. Scale bar: 2 μm, 10 s.

D   Quantification of KIF17-coupled peroxisome motility inside and outside actin patches. The average speed of KIF17-bound peroxisomes was not significantly different inside actin patches. As a control, randomization of the actin patch locations ("offset", as shown in Appendix Fig S2) was used. Two-tailed paired Student's $t$-test. $n$ = 17 dendritic segments of 12 cells in three independent cultures.

E   Representative kymograph from a DIV16 hippocampal neuron expressing PEX-GFP-KIF17 before and after LatA treatment (5 μM) (see also Fig EV4C–E). Scale bar: 5 μm, 15 s.

F   Quantification of KIF17-coupled peroxisome motility from kymographs, as in (E). Analyzed was the total time spent pausing (< 1 min), stationary (≥ 1 min), or moving in the anterograde or retrograde direction. LatA treatment (5 μM) did not significantly affect motility. RM-2-ANOVA with PEX behavior and treatment as within-group factors. $F_{(3, 63)}$ = 1.9052, $P$ = 0.138. $n$ = 22 dendritic segments of 15 cells in three independent cultures.

G   Summed total pausing time of KIF17-coupled peroxisomes in LatA (5 μM)-treated neurons, relative to the before-treatment control. LatA treatment significantly reduced the cumulative pausing time. One-sample $t$-test against 100%. **$P$ = 0.0013. Same $n$ as in (F). Data are presented as mean ± SEM.

H   Cumulative frequency of the duration of pausing events of KIF17-coupled peroxisomes before (control) and after LatA treatment (5 μM). LatA treatment did not have a significant effect on the distribution of pausing events. Two-tailed Mann–Whitney $U$-test. $n$ = 695 (control) and $n$ = 579 (LatA) pausing events in 22 dendritic segments of 15 cells in three independent cultures.

I, J   Analysis of the instant velocity and the instant run lengths of KIF17-coupled peroxisomes. LatA treatment (5 μM) decreased the average instant velocity in the anterograde direction and reduced the instant run length in the retrograde direction. Two-tailed paired Student's $t$-test. **$P$ = 0.0012 (I) and two-tailed Wilcoxon matched pairs test. *$P$ = 0.01 (J). $n$ = 22 (anterograde) and $n$ = 21 (retrograde) dendritic segments of 15 cells in three independent cultures.

pauses (Fig 7H). The total number of mobile and stationary lysosomes, as well as instant velocity and instant run length, were not affected (Fig 7F, Appendix Fig S3A and B). In a complementary set of experiments, we used a pharmacological inhibitor of myosin V, MyoVin, which partially inhibits the actin-activated ATPase activity of myosin V, thus turning it from a motor into a tether (Gramlich & Klyachko, 2017; Heissler *et al*, 2017). 30 min of MyoVin treatment (30 μM) increased the number of stationary lysosomes, the total time of long pausing events (> 1 min), and the summed pausing time (Fig EV5A–E), while the total time of short pauses (< 1 min) and instant velocity decreased (Fig EV5F). In general, we interpret these effects as MyoVin treatment leading to an elongation of the pausing time of lysosomes that are already stalled. The results from MyoV DN and MyoVin together suggest that myosin V can elongate, but not initiate the pausing time of lysosomes.

Next, we asked whether myosin VI might also be involved in the stalling of lysosomes in dendrites. To test this, we used a dominant negative construct, analogous to myoV DN, consisting of the C-terminal cargo-binding domain of myosin VI (myoVI DN) fused to GFP (Aschenbrenner *et al*, 2003; preprint: González-Gallego *et al*, 2019). Kymograph analysis of lysosomal motility in myoVI DN expressing neurons showed only a minimal change in pausing time distribution (Fig 7L) while all other analyzed parameters were unaffected (Fig 7I–K; Appendix Fig S3C and D). We subsequently employed a pharmacological myosin VI inhibitor, 2,4,6-triiodophenol (TIP) (Heissler *et al*, 2012), to test whether interference with myosin VI actin-activated ATPase activity can affect pausing of lysosomes. We found that TIP treatment slightly increased the fraction of long pauses (> 1 min) and reduced instant velocities and the run length (Fig EV5G, I and L). The pausing behavior, analyzed by summed pausing time as well as relative frequency distribution, was not changed (Fig EV5J and K). We therefore conclude that myosin VI contribution to lysosome stalling is negligible.

All in all, we found that a multitude of factors, both passive and active, are involved in lysosome positioning at the actin patches. The effects of LatA on KIF17-coupled peroxisome transport suggest that the presence of a dense actin mesh constitutes a passive, physical barrier for kinesin-mediated trafficking. In the presence of actin patches, peroxisomes, which are not associated with myosin V, take frequent but short pauses. Lysosomes, which do contain myosin V, decrease their stalling time upon inhibition of myosin V, and increase their pausing time when myosin V is blocked in F-actin-bound position. From this, we conclude that actin patches act as a physical hindrance that initially causes cargo to stall indiscriminately. This short stop might be sufficient for other factors to initiate active anchoring and longer stallings. It appears that myosin V but not myosin VI activity allows lysosomes to stop over longer periods of time, but neither of these myosins alone seem to be absolutely essential for the stalling of lysosomes in dendrites.

## Discussion

In this study, we aimed to obtain a better insight into the factors that define organelle localization in dendrites. We applied STED nanoscopy, and pharmacological and cargo trafficking assays to investigate the structural organization of the dendritic cytoskeleton, and asked how its architecture influences the trafficking of organelles using the example of lysosomes as vesicular organelles.

In differentiating neurons, small actin patches present in dendrites serve as places for preferred filopodia outgrowth (Korobova & Svitkina, 2010). Here, we investigated the presence and function of actin patches in dendrites of mature neurons. We found that the majority was spread over the dendritic shaft and is positive for excitatory but not inhibitory synaptic markers, and that their size correlates with the presence of homer1. In comparison with

spine synapses, the role and organization of the actin cytoskeleton at shaft synapses have not yet been addressed (Bosch *et al*, 2014; Konietzny *et al*, 2017; Mikhaylova *et al*, 2018). Using pharmacological approaches, we found that the actin mesh consisted of both branched and longitudinal filaments. The nucleation and maintenance of the actin patches were independent from endosomes, which makes them distinct from the axonal actin hotspots (Ganguly *et al*, 2015). The actin turnover rates at the dendritic patches are very similar to those at the dendritic spines. These observations strongly suggest that the dendritic actin patches are part of excitatory shaft synapses. Moreover, these shaft synapses presumably contribute to synaptic signaling as they are opposed to active presynaptic terminals, and are positive for both AMPA and NMDA receptors.

Dendritic microtubules shape around these dendritic actin patches, but do not invade. We therefore speculate that actin patches form strong physical barriers for newly growing microtubules, forcing them to go around. Along the same lines, the rigidity of actin patches at the base of spines could contribute to redirect growing microtubules into spines (Schätzle *et al*, 2018). Interestingly, we noticed that in contrast to F-actin patches, MT bundles were closely associated with the periodic F-actin lattice in dendrites.

Indeed, it has been shown that at least in the axon, periodic actin structures are required to maintain microtubules (Qu *et al*, 2017). That points to a very different nature and molecular environment of those two types of F-actin structures.

In another recent work, we used quantum dots to study the heterogeneity of the cytoplasm and demonstrated that the F-actin-rich cellular cortex at the inner face of the plasma membrane is able to trap diffusive probes of a certain size (Katrukha *et al*, 2017). The presence of a dense actin mesh in dendrites might therefore influence the diffusion of molecules by confining local environments, and directly or indirectly influence organelle transport. An interesting question is how vesicular organelles such as lysosomes behave in actin-rich areas. In recent years, several studies have demonstrated a role of lysosomes in synaptic plasticity (Goo *et al*, 2017; Padamsey *et al*, 2017). In addition to the classical degradation function, lysosomes can release their content into the extracellular space via $Ca^{2+}$-dependent exocytosis, which is an important mechanism allowing remodeling of the extracellular matrix in proximity to activated spines (Padamsey *et al*, 2017). Such a mechanism would require active control of lysosomal localization within dendrites, which could also apply to other vesicular organelles.

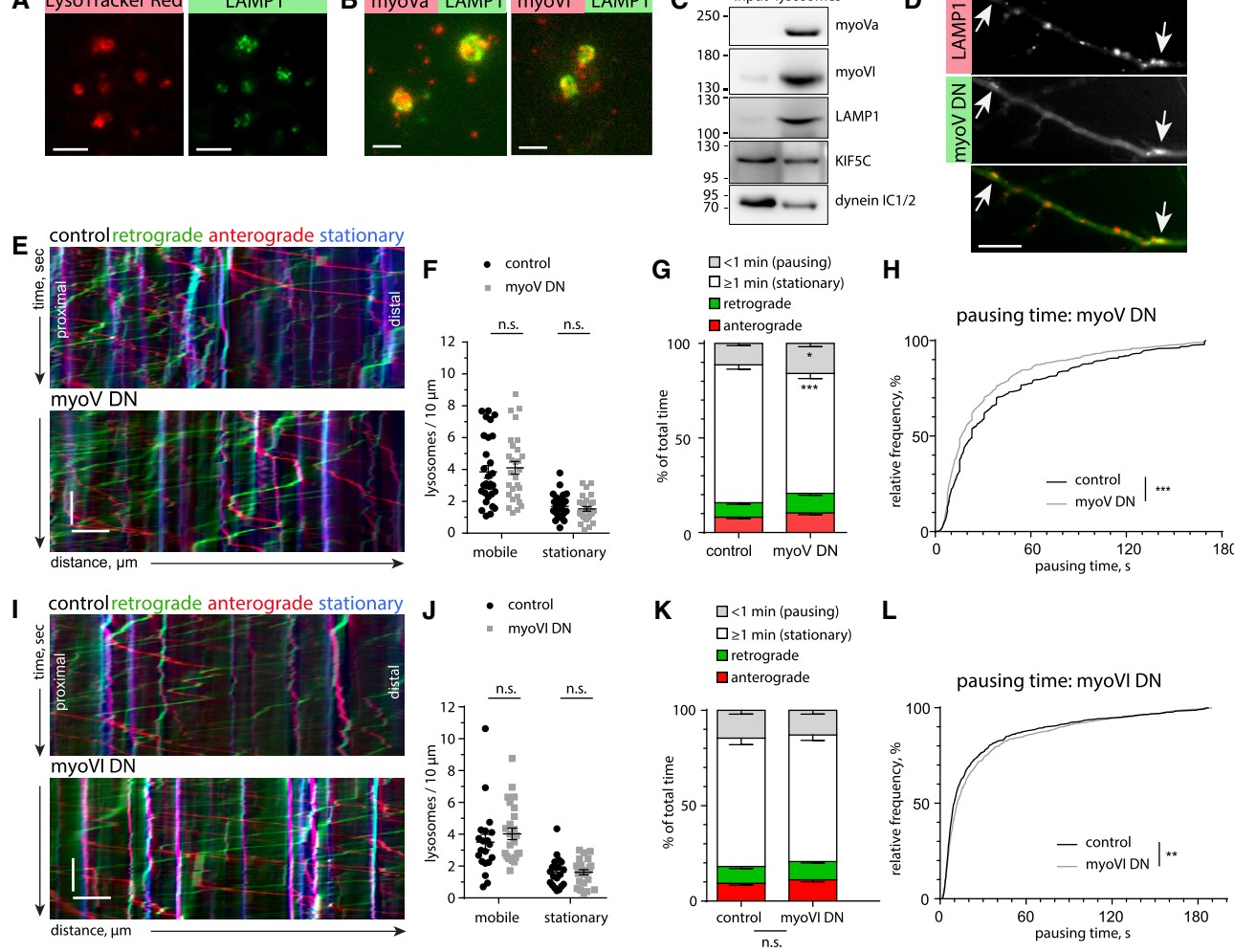

**Figure 7.**

**Figure 7. Processive myosins associate with lysosomes and myosin V affects motility of lysosomes in dendrites.**

A, B   Representative TIRF images of lysosomes enriched from adult rat cortex and stained with LysoTracker Red and α-LAMP1 antibody (A, scale bar: 5 μm), and α-LAMP1 and α-myosin Va/myosin VI antibody (B, Scale bar: 2 μm).

C   Western blot of cleared cortical lysate (input) and enriched lysosome fraction stained with antibodies against myosin Va (myoVa), myosin VI (myoVI), LAMP1, KIF5C, and dynein IC 1/2 shows the presence of different motor proteins in the lysosome fraction.

D   Dendritic segment of a DIV16 primary neuron transfected with myoV DN-mCerulean and LAMP1-mCherry. Arrows indicate myoV DN enriched at LAMP1 vesicles. Scale bar: 10 μm.

E   Representative kymographs from dendritic segments of DIV16 primary hippocampal neurons expressing LAMP1-mCherry and either mCerulean (control) or a myoV DN-mCerulean. Scale bar: 5 μm, 15 s.

F   Quantification of mobile and stationary lysosomes. Overexpression of myoV DN did not significantly change lysosome count. Unpaired two-tailed Student's *t*-test (mobile) or Mann–Whitney *U*-test (stationary). $n$ = 30 (ctr) and $n$ = 28 (DN) analyzed dendritic segments from 21 cells in three independent cultures. Data are presented as mean ± SEM.

G   Quantification of lysosome motility from kymographs as shown in (E). Analyzed was the total time spent pausing (< 1 min), stationary (≥ 1 min), or moving in the anterograde or retrograde direction. Overexpression of myoV DN led to an increase in pausing events and decreased the number of stationary lysosomes. RM-2-ANOVA with behavior of lysosomes as within-group factor and myoV DN as categorical factor. $F_{(3, 174)}$ = 6.1977, $P$ = 0.0005 with Newman–Keuls post hoc test: *$P$ < 0.05, ***$P$ = 0.0003. Control: $n$ = 31 dendritic segments of 21 cells in three independent experiments. MyoV DN: $n$ = 29 analyzed dendritic segments from 21 cells in three independent cultures. Data are presented as mean ± SEM.

H   Cumulative frequency of the duration of pausing events of lysosomes (LAMP1-mCherry) in control (mCerulean) and myoV DN-mCerulean expressing neurons. Fully stationary events are excluded. Presence of myoV DN led to a shift toward shorter pausing events. Two-tailed Mann–Whitney *U*-test. ***$P$ < 0.001. $n$ = 836 (control) pausing events in 31 dendritic segments of 21 cells, and $n$ = 973 (myoV DN) pausing events in 29 dendritic segments of 21 cells in three independent cultures.

I   Representative kymographs from dendritic segments of DIV16/DIV17 primary hippocampal neurons expressing LAMP1-mCherry and either YFP (control) or a myoVI DN-GFP construct. Scale bar: 5 μm, 15 s.

J   Quantification of mobile and stationary lysosomes. Overexpression of myoVI DN did not have a significant effect on lysosome count. Two-tailed Mann–Whitney *U*-test. $n$ = 20 (ctr) and $n$ = 25 (DN) dendritic segments of 20 cells (control) and 23 cells (myoVI DN) in two independent cultures. Data are presented as mean ± SEM.

K   Quantification of lysosome motility from kymographs as shown in (I). Analyzed was the total time spent pausing (< 1 min), stationary (≥ 1 min), or moving in the anterograde or retrograde direction. Overexpression of myoVI DN did not cause significant changes. RM-2-ANOVA with behavior of lysosomes as within-group factor and myoVI DN as categorical factor. $F_{(3, 129)}$ = 0.25, $P$ = 0.86. Same $n$ as in (J). Data are presented as mean ± SEM.

L   Cumulative frequency of the duration of pausing events of lysosomes (LAMP1-mCherry) in control (YFP) and myoVI DN-GFP-expressing neurons. Fully stationary events are excluded. Presence of myoVI DN led to a shift toward longer pausing events. Two-tailed Mann–Whitney *U*-test. **$P$ = 0.009. $n$ = 838 (control) in 20 dendritic segments of 20 cells, and $n$ = 853 (MyoVI DN) pausing events in 25 dendritic segments of 23 cells in two independent cultures.

Here, we used LysoTracker and LAMP1 to label lysosome-related compartments and observed that the labeled vesicles could be divided into stationary and mobile fractions, with the latter being moved bidirectionally in dendrites. STED imaging indicated that lysosomes can be trapped deep inside actin patches or be localized near the borders. Similarly to Goo *et al*, we found that treatment with the F-actin-depolymerizing agent LatA reduced the fraction of stationary lysosomes (Goo *et al*, 2017). Interestingly, at basal conditions there was some bias toward anterograde trafficking of lysosomes, which was diminished upon LatA treatment. This effect could be due to differential effects of the F-actin mesh on cargoes powered by different types of microtubule motors. Moreover, while performing live imaging experiments, we found that lysosomes frequently stop at actin patches. Retention of organelles at actin patches can be passive, as they form physical barriers within dendrites, or an active process, in which myosin motors or potentially other actin-binding proteins bind to the F-actin. In addition, a complex interplay between the activities of different MT motors attached to the same vesicle could affect cargo transport in proximity to F-actin patches.

To test whether the F-actin mesh at the shaft synapse could serve as a passive organelle trap, we designed a cargo-motor controlled assay, in which usually immobile peroxisomes were used as an artificial cargo for a constitutively active KIF17 motor. This organelle probe allowed us to rule out a "tug of war" between different types of MT motors attached to the same cargo. Analysis of peroxisomal movement in dendrites demonstrated that, at least to some extent, purely kinesin-driven organelles can be stalled at F-actin-rich areas. This indicates that the F-actin mesh can be an obstacle for MT motor-driven vesicular organelle transport.

Next, we tested a potential contribution of the processive motors myosins V and VI in lysosome stalling. They are known to mediate cargo transport into and inside dendritic spines (Correia *et al*, 2008; Esteves da Silva *et al*, 2015), and dendritic cargo sorting at the axon initial segment is based on the presence of myosin V (Janssen *et al*, 2017). Thus, it is very likely that many dendritic cargos are associated with myosins. Enriched lysosomes from rat brains, indeed, appeared to associate with myosins Va and VI. We explored their contribution to pausing events of lysosomes by combining dominant negative forms of myosin and pharmacological approaches. Dominant negative inhibition of myosin V led to a reduction in stalling time, whereas inhibition of myosin VI had very little to no effect. We therefore speculate that within dendrites the trafficking of endolysosomal/vesicular organelles is affected by the presence of F-actin patches, via both passive stalling and activation of specific actin-binding proteins, including myosin V.

The $Ca^{2+}$/calmodulin dependency of myosin V activation could provide a mechanism for activity-dependent positioning of lysosomes (Wang *et al*, 2008; Goo *et al*, 2017). It is conceivable that only a subset of lysosomes will carry myosins, possibly depending on their maturation status. In a recent study, Cheng and colleagues performed a rigorous characterization of LAMP1-positive compartments in dendrites. They found that about half of the LAMP1-labeled structures contained functional lysosomal proteases cathepsins B and D, markers for mature lysosomes (Cheng *et al*, 2018). It has also been shown that neuronal activity induces lysosomal exocytosis and release of cathepsins (Padamsey *et al*, 2017). The spatially and temporally controlled cathepsin B release from mature lysosomes may be a critical step required for input-specific synaptic potentiation. The close proximity to NMDA receptors at

shaft synapses would allow for direct $Ca^{2+}$-dependent synaptic control of lysosomal exocytosis. A dense actin mesh surrounding the PSD of shaft synapses or at the base of spines could stall mature fusion-competent lysosomes for longer periods. Alternatively, long pausing events might promote endolysosomal maturation. How trafficking rules relate to lysosomal maturation needs to be addressed in future studies.

The finding that actin patches are part of active excitatory shaft synapses raises many new exiting questions. In contrast to spine synapses, the organization, stability, and function of excitatory shaft synapses remain largely unexplored. Some studies indicate that during neuronal development, shaft synapses are precursors for spine synapses, while others suggest that they comprise an independent category of synapses, also in the mature brain (Aoto *et al*, 2007; Bourne & Harris, 2011; Reilly *et al*, 2011). It is interesting to know whether and how activation of shaft synapses influences organelle transport. Opening of NMDA receptors at shaft synapses will lead to direct influx of calcium into the dendritic shaft. Shaft synapses might therefore contribute more strongly to controlling organelle trafficking in comparison with spine synapses, since they lack the spatial confinement by a spine neck (Sabatini *et al*, 2002). We anticipate that dense dendritic actin patches could also limit molecule diffusion and encourage local confinement. Calcium influx at a shaft synapse could result in local dendritic areas with changed traffic signals, allowing for coordinated cargo delivery and positioning in places of high demand. Moreover, the size and density of the F-actin mesh at shaft synapses or actin patches at the spine base could be rapidly altered by synaptic activation, providing a fine-tuning of its size and complexity.

In summary, dendritic trafficking of organelles is complex and involves many parameters including the architecture of the cytoskeleton, associated motor proteins, and synaptic activity. Here, we looked deeply into the cytoskeletal architecture using super-resolution microscopy and live cell imaging techniques. The results illustrate that dendritic actin patches, at spine bases and at shaft synapses, can influence organelle trafficking. Whether these F-actin patches could serve as a hub for bringing organelles in proximity to each other, promoting organelle–organelle contact, and how this relates to organelle functioning and maturation, e.g., during plasticity events, deserves further investigation.

# Materials and Methods

## Animals

Wistar rats Crl:WI (Han) (Charles River) and Wistar Unilever HsdCpb:WU (Envigo) rats were used in this study. Sacrificing of pregnant rats (E18) for primary hippocampal cultures, P7 male rat pups for organotypic slice cultures, and adult female rats for biochemistry was done in accordance with the Animal Welfare Law of the Federal Republic of Germany (Tierschutzgesetz der Bundesrepublik Deutschland, TierSchG) and with the approval of local authorities of the city-state Hamburg (Behörde für Gesundheit und Verbraucherschutz, Fachbereich Veterinärwesen, from 21.04.2015) and the animal care committee of the University Medical Center Hamburg-Eppendorf.

## Hippocampal neuronal primary cultures, transfections

Rat primary hippocampal cultures were prepared as described previously with slight modifications (Kapitein *et al*, 2010). In short, hippocampi were dissected from E18 embryos, physically dissociated after 10 min of trypsin treatment (0.25%, Thermo Fisher Scientific, #25200-056) at 37°C, and plated on poly-L-lysine (Sigma-Aldrich, #P2636)-coated glass coverslips (18 mm) at a density of 40,000–60,000 cells per 1 ml on in DMEM (Gibco, #41966-029) supplemented with 10% fetal calf serum (Gibco, 10270) and antibiotics (Thermo Fisher Scientific, #15140122). After 1 h, plating medium was replaced by BrainPhys neuronal medium supplemented with SM1 (Stem Cell Kit, #5792) and 0.5 mM glutamine (Thermo Fisher Scientific, #25030024). Cells were grown in an incubator at 37°C, 5% $CO_2$, and 95% humidity.

Cultures were transfected at indicated time points using lipofectamine 2000 (Thermo Fisher Scientific, #11668019). The DNA/lipofectamine ratio was according to the manufacturer's protocol 1:2. For co-transfection of plasmids, the ratios of different constructs were optimized per combination, and optionally by addition of an empty vector (pcDNA3.1), to tune expression levels. Before transfection, the original neuronal medium was removed. Neurons were transfected in BrainPhys medium, optionally supplemented with glutamine, but lacking SM1. Transfection medium was added for 45 min–1.5 h. After transfection, the medium was exchanged back to the original BrainPhys containing SM1. Experiments on transfected neurons were performed 1 day after transfection.

## Immunocytochemistry

Primary cultures were fixed with 4% Roti-Histofix/4% sucrose for 10 min at room temperature (RT), washed 3 × 10 min with phosphate-buffered saline (PBS), and permeabilized with 0.2% Triton X-100 in PBS for 10 min at RT. The cells were then washed 3× in PBS and blocked for 45 min at RT with blocking buffer (10% horse serum, 0.1% Triton X-100 in PBS). Primary antibodies were added in blocking buffer and incubated overnight at 4°C. Cells were washed 3 × 10 min before addition of secondary antibodies in blocking buffer, and incubation for 1 h at RT. Optionally, phalloidin-Atto647N was added 1:40 in PBS and incubated overnight at 4°C. Finally, coverslips were washed 3–5 × 10 min in PBS and mounted on microscope slides with Mowiol. Mowiol was prepared according to the manufacturer's protocol (9.6 g Mowiol 4-88 (Carl-Roth), 24.0 g glycerin, 24 ml $H_2O$, 48 ml 0.2 M Tris pH 8.5, including 2.5 g Dabco, (Sigma-Aldrich D27802)).

## Live cell imaging: wide-field, TIRF, and spinning-disk microscopy

Live cell wide-field and TIRF microscopy was performed with a Nikon Eclipse Ti-E controlled by VisiView software (Visitron Systems). Samples were kept in focus with the built-in Nikon perfect focus system. Fluorophores were excited by 488, 561 and 639 nm laser lines, coupled to the microscope via an optic fiber. HILO and TIRF illuminations were obtained with an ILAS2 (Gattaca systems) spinning-TIRF system. Samples were imaged with a 100x TIRF objective (Nikon, ApoTIRF 100×/1.49 oil). Emission light was collected through a quad-band filter (Chroma, 405/488/561/640) followed by a filter wheel with filters for GFP (Chroma, 525/50 m),

RFP (Chroma, 595/50 m), and Cy5 (Chroma, 700/75 m). Multichannel images were acquired sequentially with an Orca flash 4.0LT CMOS camera (Hamamatsu). Images were acquired at 2–5 frames per second or at specified intervals.

FRAP of FusionRed-actin was performed using the ILAS2 unit. FusionRed was imaged and bleached using a 561 nm laser. Photobleaching was performed with 2 ms per pixel, with an approximate twofold laser intensity. Images were acquired with 3 s of increments in HILO illumination, starting with 10 frames of baseline before FRAP.

Spinning-disk confocal microscopy was performed with a Nikon Eclipse Ti-E controlled by VisiView software. Samples were kept in focus with the built-in Nikon perfect focus system. The system was equipped with a 100× TIRF objective (Nikon, ApoTIRF 100×/1.49 oil), and 488, 561 and 639 nm excitation laser. Lasers were coupled to a CSU-X1 spinning-disk (Yokogawa) unit via a single-mode fiber. Emission was collected through a quad-band filter (Chroma, ZET 405/488/561/647m) on an Orca flash 4.0LT CMOS camera (Hamamatsu). Images were acquired sequentially with 0.3–3 frames per second or at specified intervals.

Additional spinning-disk confocal microscopy was performed with a Nikon Eclipse Ti-E controlled by VisiView software. Samples were kept in focus with the built-in Nikon perfect focus system. 488 and 561 nm excitation lasers were coupled to a CSU-W1 spinning-disk unit (Yokogawa) via a single-mode fiber. Cells were imaged with either a 60× (Nikon, ApoTIRF 60×/1.49) or a 100× (Nikon, CFI Plan Apochromat Lambda 100×/1.45) objective. Emission light was split by a 561 LP dichroic and filtered through a GFP (Chroma, 525/50 m), or a mCherry filter (Chroma, 609/34m). Multichannel images were acquired simultaneously on two EM-CCD cameras (Photometrics, Evolve 512 Delta). Images were acquired at 4 frames per second.

At all systems, neurons were imaged in regular culture medium. Coverslips were placed in either an attofluor cell chamber (Thermo Fisher Scientific) or a Ludin chamber (Life Imaging Services). Correct temperature (37°C), $CO_2$ (5%), and humidity (90%) were maintained with a top stage incubator and an additional objective heater (Okolab).

If pharmacological treatment was performed during live imaging, the respective drug was added manually to the culture medium and incubated for the indicated time spans in the top stage incubator.

## Confocal, gatedSTED imaging, and deconvolution

Confocal microscopy of fixed primary cultures was performed at a Leica TCS SP5 confocal microscope (Leica Microsystems, Manheim, Germany). The microscope was controlled by Leica Application Suite Advanced Fluorescence software. Samples were imaged using a 63× oil objective (*Leica*, HC PL APO CS2 63×/1.40 oil). Fluorophores were excited with a 488 nm Argon laser, 561 nm diode-pumped solid-state laser, and a 633 nm He-Ne laser. Images were acquired at 1,024 × 1,024, with a 60 nm pixel size, in 8 bit. To reduce noise, a two times frame averaging was applied. For z-stacks, z-step size was set to 0.29 μm.

Fixed organotypic slices were imaged at an Olympus FV1000 confocal microscope with a 40× objective (Olympus, UPLFLN 40×/1.3). Fluorophores were excited by a 488 nm and a 559 nm laser line. Pinhole size was set to 80 μm. Excitation and emission light

was divided by a quad-band filter (405/488/559/635). Emission light was split by a 570b nm dichroic and filtered through a GFP (495–540 nm) or a RFP (575–630 nm) filter. Images were acquired sequentially with two high sensitivity detectors. Pixel size was 77 nm, with a z-step size of 0.2 μm. Single planes consisted of 3 frame averages, 12-bit.

gatedSTED images were acquired at a Leica TCS SP8-3X gatedSTED system (Leica Microsystems) equipped with a pulsed white light laser (WLL) for excitation ranging from 470 to 670 nm. STED was obtained with a 592 nm continuous wave and a 775 nm pulsed depletion laser. Samples were imaged with either a 100× objective (Leica, HC APO CS2 100×/1.40 oil) or a 93× glycerol objective (Leica, HC APO 93×/1.30 GLYC motCORR). The refractive index (RI) of Mowiol (see immunocytochemistry) polymerized for 3 days was 1.46 (measured with Digital Refractometer AR200 (Reichert), and matched the RI of glycerol (1.45) better than oil (1.51). Therefore, when available, the 93× glycerol objective was the preferred objective. For excitation of the respective channels, the WLL was set to 650 nm for phalloidin-Atto647N, 561 nm for Abberior STAR 580, and 488 nm for either Alexa Fluor 488-conjugated secondary antibodies or GFP-fusion proteins. STED was attained with the 775 nm laser for Atto647N/Abberior 580 and with the 592 nm laser for Alexa Fluor 488/eGFP. Emission spectra were detected at 660–730 nm for Atto647N, 580-620 nm for Abberior STAR 580, and 500–530 nm for Alexa Fluor 488 or eGFP. For gatedSTED, detector time gates were set to 0.5–6 ns for Abberior STAR 580/Atto647N and 1.5–6 ns for Alexa Fluor 488/eGFP. Images were acquired as single planes of either 1,024 × 1,024 pixels or 1,386 × 1,386, optical zoom factor 5 (for oil: x/y 22.73 nm, for glycerol: x/y 24.44 nm or 18.28 nm) or 6 (glycerol x/y 20.37 nm), 600 lines per second, and 16× line averaging. Corresponding confocal channels had the same setting as STED channels, except the excitation power was reduced and the detection time gates were set to 300 ps – 6 ns for all channels.

Deconvolution of STED and confocal laser scanning (CLSM) images was done with Huygens Professional (Scientific Volume Imaging). Within the deconvolution wizard, images were subjected to a background correction, and signal-to-noise ratio was set to 15 for STED and 30 or 40 for CLSM images. The optimized iteration mode of the CMLE was applied until it reached a quality threshold of 0.01 for STED and 0.05 for CLSM images.

## Kymograph analysis

Kymographs were constructed using the KymographClear (Mangeol *et al*, 2016) or KymoResliceWide plugin for Fiji (NIH) (Schindelin *et al*, 2012). Non-overlapping dendritic stretches of 30–80 μm length were traced, using the segmented line tool. Maximum 3 dendrites were taken from the same neuron. Line thickness was chosen to cover the dendritic shaft, but to omit spines. Trajectories in the kymographs were traced by hand using the straight line tool. Trajectories with a uniform speed (= slope) were considered one "event", and changes in speed (= slope) or stopping events (= vertical lines) were traced as separate events (Appendix Fig S1). Slope and length of each event were used to calculate instant velocity, instant run length, and pausing time. In general, all lysosomes were analyzed. For experiments with the PEX3-Kif17 construct, only peroxisomes that showed at least one moving behavior were analyzed.

Non-moving peroxisomes, potentially due to inefficient coupling, were thereby excluded.

### Data representation and statistical analysis

For representative microscopy images, brightness and contrast are linearly adjusted per channel. Statistical analysis was performed in Statistica 13 (Dell Inc.) or Prism 6.05 (GraphPad). Detailed specifications about the type of test, significance levels, n numbers, and biological replicates are provided in the figure legends. Data are represented individually in dot blots or as mean ± SEM throughout the manuscript.

Expanded View for this article is available online.

### Acknowledgements

The authors would like to thank W. Wagner (*MyosinV-DN_mCerulean*), A.S. Kostyukova (FusionRed-actin), and T.G. Oertner (mCerulean, mRuby2) for sharing plasmids. Psd95.FingR_eGFP_CCR5TC was a gift from Don Arnold (Addgene plasmid #46295). We thank UMIF for access and use of their spinning-disk and confocal microscopes. This work was supported by grants from the Deutsche Forschungsgemeinschaft (DFG): Emmy-Noether Program (MI 1923/1-1), FOR2419 (MI 1923/2-1 and MI 1923/2-2), and CRC877 (INST 257/602-1) to M.M. and DFG grant SCHE 132/18-1 to O.K.

### Author contributions

BvB, AK, JB, and MM designed the study. BvB, AK, OK, JB, and MM conducted the experiments. BvB, AK, JB, and MM analyzed the data. MM supervised the project and wrote the manuscript. All authors commented on and revised the manuscript.

### Conflict of interest

The authors declare that they have no conflict of interest.

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
