## [Review Process File · The EMBO Journal]

F-actin patches associated with glutamatergic synapses control positioning of dendritic lysosomes

Bas van Bommel, Anja Konietzny, Oliver Kobler, Julia Bär, Marina Mikhaylova

Review timeline:

Submission date:	15th Nov 2018
Editorial Decision:	20th Dec 2018
Revision received:	30th Apr 2019
Editorial Decision:	22nd May 2019
Revision received:	27th May 2019
Accepted:	29th May 2019

Editor: Karin Dumstrei

Transaction Report:

1st Editorial Decision

20th Dec 2018

Thanks for submitting your manuscript to The EMBO Journal. Your study has now been seen by three referees and their comments are provided below.

As you can see from the comments below, the referees find the analysis interesting and suitable for publication here. However, they also raise a number of relevant points that should be addressed. I anticipate that you should be able to address most of them, but let me know if we need to discuss some of them in more detail.

Given the referees' positive recommendations, I would like to invite you to submit a revised version of the manuscript, addressing the comments of all three reviewers.

REFeree REPORTS:

Referee #1:

In this study, van Bommel and colleagues show that actin patches are present along the dendritic shaft, sometimes at the base of spines and often apposed to shaft glutamatergic synapses. They provide splendid - but a bit small, see below - images of these actin assemblies and characterize them in detail in the first part of the manuscript. In the second part, they explore the function of these actin structures in regulating the transport of dendritic organelles, in particular lysosomes. The findings are very topical and interesting, as super-resolution microscopy is currently redefining the organization of neuronal actin in mature neurons (rings along the axon and dendrites, axonal patches, hotspots and trails...). As usual for good studies, this works opens up to many interesting questions. Even if I can't help but wish some of them were addressed straight away, I will try to focus on the current perimeter of the study, namely the nature and organization of these actin patches, and their role in the regulation of axonal transport. I think a number of points should be addressed in a revised version of this manuscript before being able to recommend acceptance for publication in EMBO Journal.

Major points

1- Patches characterization - spine base-associated vs shaft synapse-associated patches

The two types of patches are described as being similar, but they could be two distinct structures with different partners and function, as spine-related patches seem to not be associated with synapses (as stated in the Results). The authors should be careful and try to compare the two to distinguish them. Spine base associated patches should be close to the plasma membrane; are the shaft synapse-associated patches located close to the membrane (around synapses) or deeper in the dendrite, below the synaptic contact? Are the results on lysosome and PEX3-KIF17 transport from Fig. 6 similar for spine-associated patches and shaft synapse-associated patches?

2- Patches characterization - images

In the first part characterizing the actin dendritic patches, the STED images are beautiful, but often too small. It is good to have a large overview of a substantial segment of dendrite, but zoom boxes would help distinguishing the small features and local organization of actin patches. A good example of such zoomed regions is Fig. 4B - it would be good to generalize this to several other Figures describing the organization and partners of actin patches. For Figure 4B, the Results text says "LAMP1 puncta were distributed all over the dendrite" but these puncta are very difficult to see on the low-mag overlay image - perhaps panels with separated channels like on other Figures would help. Lastly, the 3D reconstruction from Fig. EV2B is surprising: in the view presented, all the synapses that appear over the shaft are "shaft synapses" (red). In a slice where environment is isotropic (no substrate effect), I would expect "spine synapses" (green) to be randomly oriented, and some of them to go upward and appear over the shaft on the view, rather than appearing all on either side, distant from the shaft. Could the authors present an additional orthogonal view of the same dendritic segment to clarify this?

3- Patches characterization - actin perturbations and cortactin presence

The Figure 3 and EV3 are showing the effect of actin-perturbing drugs on dendritic actin patches. The doses and treatment duration for each drug should be stated each time in the main Text/Figure/Legends (one or the other), rather than just as a table in the supplementary material. The effects are qualitatively reported as a more or less "reduction of F-actin at patches" - would it be possible to quantify this to make it more objective? Similarly, the colocalization of cortactin with actin patches should be quantified: what % of actin patches are cortactin-enriched? In line with point 1, is there a difference between spine base-associated and shaft synapse-associated patches for the effect of actin-perturbing drugs, and for cortactin enrichment (notably in relation to the cortactin enrichment at the base of spines described by Schätzle et al. *Curr Biol* 2018)?

4- Transport experiments - Generalization

Would it be possible to extend the results about the effect of actin patches on dendritic transport to other organelles or dendritic cargoes, beyond lysosomes and synthetic mobile peroxisomes, in order to support a general role? What happens for mitochondria?

5- Transport experiments - Streamlining of results

- The functional part of showing the effect of actin patches on the dendritic transport of organelles is interesting. However, in order to be more convincing, the results should be presented in a more simple and uniform way, without diluting the eventual differences in a large number of measured parameters. For example, for lysosomes, no effect is found for latA on the average velocity (Fig. 5D and 5I), but the speed is different in actin-rich regions (Fig. 6A). For PEX3-KIF17, by contrast, there is no difference in speed in actin-rich regions (Fig. 6C), but latA has a significant effect on the average velocity (Fig. 6E). The authors nonetheless state that the latA experiment performed on PEX3-KIF17 "address this apparent controversy" and "suggests, similar to lysosomes" [that have no difference in velocity after latA treatment] "an interaction with the actin cytoskeleton". Also, the PEX3-KIF17 data on the effect of LatA on velocity comes from a single experiment - replicates should be obtained.

- Quantification presented should be performed and presented in the same way across experiments. In particular, the quantification of pauses is quite confusing because it changes depending on the experiment and Figure. It is very difficult to assess and interpret what happens as the data are never presented with the exact same set of visualizations. Could the author provide the same set of quantification and visualization for all experiments: % of total time pausing/stationary/retrograde/anterograde (shown in Fig. 5B, 5G, 7F, 7I, EV5B, EV5F but not for

PEF-Kif17 in Fig.6); % of pausing events (as in Fig. 7F, 7I, EV5B, EV5F but not for ctrl/latA in Fig.5 or PEX3-KIF17 in Fig. 6); number of immobile/mobile lysosomes per 10 μm (as in Fig. 7G, 7I but not for ctrl/latA in Fig.5, PEX3-KIF17 in Fig. 6 or myosin inhibitors on Fig. EV5); cumulative frequency of pausing times (shown in Fig.6G for PEX3-KIF17 but not Fig.5 for ctrl/latA, shown in Fig. EV5C and EV5G for myosin inhibitors but not in Fig.7 for myosin dominant negatives). Finally, I don't think the unique and odd "normalized pausing time" shown for PEX3-KIF17 in Fig. 6F should be kept or generalized to all relevant Figures.

6- Transport experiments - Role of myosins

- The data showing association of myosins with lysosomes is obtained on lysosomes purified from rat brain. Is it possible to show myosin presence on lysosomes in the same system as the rest of the study, i.e. in cultured neurons with immunocytochemistry and STED?
- The authors should show the effect of myosin perturbation by DNAs or drugs on the correlation between transport parameters and actin patches localization: do myosin perturbations remove the lysosomes transport modification at actin patches shown in Fig. 6A-B?
- The results from the DNAs and inhibitors are not very clear - I'm not sure if the inhibitor data supports the conclusion that myosin V is implicated, but not myosin VI as the effect shown on Fig. EV5 are consistently more significant for TIP than for myosin for most parameters.
- About the interpretation of the PEX3-KIF17 data as a demonstration of "passive" effect of actin patches: peroxisomes are immobile, but is it known that they don't have associated myosin V/VI on their surface?

Minor points

- When discussing the recently described actin structures in dendrites, it would be relevant to cite Nithianandam & Chien JCB 2018, which described actin "blobs" along dendrites that precedes dendritic branching.
- The work of Schätzle et al. (Curr Biol 2018) is incidentally cited in the Discussion, but it clearly shows the presence of actin accumulation at the base of dendritic spines, so could be cited when discussing recently described actin structures in the Introduction.
- In the Introduction, when discussing molecular processes underlying plasticity and stability of synaptic contacts, the surface diffusion of membrane components (such as ionotropic receptors) could be added.
- The transfection times are indicated throughout the text, but I could not find what was the time between the transfection and the experiments themselves. Please precise this throughout or once in the Methods section if constant for all experiments.
- How were LatA or other drugs added in the live-cell imaging of transport experiments? Was a perfusion system used or manual addition of concentrated drug?
- Results p.5, last line: "complimentary" seems to be switched for "complementary"

Referee #2:

The manuscript by Bommel et al. describes the presence of actin patches in the dendritic shafts of cultured hippocampal neurons that could act to trap mobile organelles near excitatory postsynaptic sites on the dendritic shaft. While heterogeneity in dendritic F-actin staining has been widely observed, this study seeks to characterize the potential functional role of these actin patches as organelle tethers or barricades. The study uses a combination of traditional confocal, live cell microscopy in conjunction with super resolution STED microscopy to make the case for actin patches being associated with postsynaptic densities on the dendritic shaft and the study provides some evidence for steric and active (via myosinV) trapping of mobile organelles trafficking along microtubules by the patches. The authors propose a model where actin patches could direct vesicular cargo to excitatory synapses on dendritic branches (and because they are also situated at the base of dendritic spines, they could also direct mobile organelles into spines for synaptic trafficking). While the observation of actin patches within dendrites isn't novel, the manuscript breaks new ground by demonstrating a selective association of the patches with excitatory synaptic sites and providing support for a model where mobile dendritic organelles pause at these sites. However, I have a number of concerns that would need to be addressed before I would recommend publication in EMBOJ.

Major concerns:

-Overall there was little attempt to quantify the imaging data in figures 1-4. While the sample images displayed look compelling, the authors need to add additional quantitative analyses to these figures. For example, what is the distribution of sizes of the actin patches? Does the actin patch size correlate with the size of the associated PSD? Can you estimate how close the edges of actin patches are to microtubules? The authors should use an established metric to describe how well actin patch signal correlates with various synaptic marker proteins.

-The synaptotagmin feeding experiments in Fig. 1C needs a negative control (preventing neural activity with TTX or similar during antibody exposure) to ensure the signal is specific to synaptic vesicle turnover. Also, showing only the chromobody/synaptotagmin channels here as a 2-color merge would be a cleaner way to demonstrate co-localization of functional terminals with postsynaptic actin since a significant fraction of the phalloidin signal likely comes from axons and presynaptic terminals. A separate set of images demonstrating colocalization between phalloidin and the chromobody could validate the chromobody labeling.

-Figure 2A relies on phalloidin staining to demonstrate patches colocalize with homer, but not gephyrin. I would also like to see this with an expressed marker for postsynaptic actin (e.g. sparsely expressed chromobody or lifeact) as some of the phalloidin signal could arise from presynaptic terminals.

-The authors report widely ranging values and metrics for organelle mobility. I couldn't find any information about how these were calculated. These include "average speed" (e.g. Fig. 6B), "instant velocity" (e.g. Fig. 5I), and "average instant velocity" (e.g. Fig. 5D). How were these calculated and why do they give widely varying values (~0.1 micrometer/sec, average speed Fig. 6D vs ~1.2-1.5 micrometer/sec for "average instant velocity", Fig. 6E).

-It's hard to understand how organelle velocities were selectively measured within the actin patches since these structures are so small (near diffraction-limited). How many imaging frames did the organelles take to traverse these tiny compartments? How could the authors know the organelle was in contact with the actin patch since they were using diffraction-limited microscopy?

-Treating the neurons with brefeldin A for 10h to conclude endosomes are not involved in actin patch generation seems like an indirect, and possibly misleading, experiment. I recommend omitting it.

-While the actin patches appear stable over time, the authors should provide more quantitative dynamic measurements (using FRAP and/or single molecule tracking for example). Are actin patches near spines more or less dynamic? Are the PSD associated shaft patches more/less dynamic than neighboring non-PSD associated patches?

-In Fig. EV3B, how do you know SiR-actin labels only postsynaptic actin?

Minor issues:

-There is a callout to Fig. "S2" on pg. 8; there is no Fig. S2.

-Scale bars are missing in some imaging panels (Fig 1B far right panels showing blown up images)

-On pg. 7 there is a reference to "axonal hotspot" with no context give for what this means

-It is not immediately apparent why randomizing the location of the actin patches in Fig. 6B (lower panel) would have an "opposing effect" on organelle velocity since these patches presumably occupy a relatively small fraction of the total dendritic volume.

-No reference was included for the actin "chromobody" (Fig. 1C). I assume this is the commercially available reagent from Chromotek? References and details need to be included.

Referee #3:

In this manuscript the authors propose interesting model where actin filaments act as molecular traps to impede the movement of endosomal vesicles (and perhaps help deliver these cargoes to locations within the dendrite). Though the question is an interesting one, there are significant conceptual and technical concerns that strongly limit enthusiasm for this model. Specific points are noted below.

1. A major issue with this manuscript is that it is not clear what the authors mean when they talk about "actin patches". Dendritic shafts have significant accumulations of actin, including enrichment in postsynaptic densities. The authors only provide a qualitative description without any serious attempt at clearly defining the very structures that they are describing. For instance in figure 1, there are numerous dots and patches of actin in the dendritic shaft, and the arrows only point to a few of them. Some of them look sub-plasmalemmal (and linear), while others are at the base of the spine and neck and look somewhat pyramidal in shape. Many other dots, spots and speckles are seen throughout the dendrite. In figure 1B it seems that the authors are pointing to postsynaptic densities, which are known to be enriched and actin. Indeed, the actin densities shown in the supplementary figure are clearly postsynaptic densities. How are these densities different from what the authors are calling "actin patches".....or are they the same? No quantitative statements are made that strongly limits the impact of this work.
2. The older cultures are complicated, with numerous axons and dendrites intertwining - all of which have actin in them. How can the authors be sure that what they're looking at is definitely within the shaft of the dendrite? Though one realizes that this is a difficult problem to address, the onus is still on the authors to resolve this as clearly as possible.
3. The synaptotagmin uptake shown in figure 1C is not convincing.
4. For the authors model to be true, it is very important that the actin patches are not dynamic, however this is not been shown clearly. The authors use Sir-actin, which is a Taxol based probe that stabilizes actin, and it is not enough to conclude that the patches are stable. In fact the kymographs of actin chromobody imaging shown in figure 6 suggest that these patches are dynamic. The authors should have used a more physiologic marker like Lifeact to make this point. The polymerization of the actin structures upon latA (figure 3) also strongly indicate that these accumulations are dynamic.
5. In figure 2E, there is no quantitative analysis.
6. The quality of data in figure 4 is not very good, and it is difficult to agree with the author's conclusions (that are again qualitative).
7. Data in figure 5 is interesting, showing that the transport frequency of lysosomes and dendrites is increased upon treatment with lat A, and the anterograde bias is lost. However, it is unclear if these phenotypes have anything to do with actin patches. Why would the patches be involved in abrogating the anterograde bias of lysosomes? What is the effect of actin depletion on other dendritic cargoes? (the pex experiments are not useful). The concentration, time of lat A usage (and efficacy of actin-depletion in these older neurons) is also not clear.
8. Figure 6: it is not clear that the puncta seen by the actin chromobody really represents the actin patches (especially because the patches themselves are not well defined). The kymographs shown have much less LAMP GFP movements, compared to the other kymographs shown in the previous figure. A critical unanswered question here is the frequency of vesicle movement. How many endosomes really pause at these patches? In many cases the slowing is very modest, and the statistical significance seems odd (Fig. 6B - below for example has **?).

In summary, there is some very impressive imaging in this manuscript, and the authors model is also an interesting one. The slowing of endosomal cargoes, and elimination of the bias upon actin depletion is also interesting, but it is really unclear the effect is due to the actin patches (or even what the actin patches really are). It is also unclear if the actin densities are really stable. Other points are noted above. Collectively, these concerns limit enthusiasm for this manuscript.

Referee #1:**Major points****1- Patches characterization - spine base-associated vs shaft synapse-associated patches**

The authors should be careful and try to compare the two to distinguish them. Spine base associated patches should be close to the plasma membrane; are the shaft synapse associated patches located close to the membrane (around synapses) or deeper in the dendrite, below the synaptic contact? Are the results on lysosome and PEX3-KIF17 transport from Fig. 6 similar for spine-associated patches and shaft synapse-associated patches?

Response:

We would like to thank this reviewer for very constructive comments on the manuscript. We have now provided further analyses to distinguish spine- and shaft associated patches, including comparing the sizes of the actin patches as well as presence or absence of the post-synaptic density protein homer1. We discovered some distinguishing parameters between spine- and shaft associated F-actin loci. For instance, homer1-positive patches in the shaft tend to be larger than homer1-negative patches; shaft associated patches more frequently are homer-1 positive than the spine base associated F-actin. Now these new data are included in the Figure 2F. We do not want to make strong statements about the association of patches with the plasma membrane because the data were acquired in 2D STED mode. From this we cannot draw conclusions about the localization of the actin patches in 3D space. For instance, we cannot exclude that patches that appear to be deeply localized within the dendritic volume are in fact close to the plasma membrane contacting the coverslip.

Regarding the question about possible differential effects of actin patches at the spine base vs. shaft on cargo trafficking presented in the Figure 6A-D, we cannot fully discriminate shaft vs. spine contribution because these experiments are performed on a spinning disc microscope and not in a STED system. The confocal spinning disc allows detection of patches but is not good enough to make an exact statement about patches being associated with a spine or not. At this stage of development our neuronal cultures have high spine densities (about 8-10 per 10 μm) and we did not want to overinterpret the data. However, we assume that there will be some differences between the two locations, considering their different sizes and different sensitivity to pharmacological interruption, which points at a distinctive regulation of actin dynamics / stability. On the other hand, other factors, like activation of dendritic spine or a shaft synapses could lead to a local increase in calcium concentration and influence the actin dynamics of the patch as well as lead to the activation of myosin V and other actin-binding proteins. We are currently exploring these options in a follow up project.

2- Patches characterization – images

In the first part characterizing the actin dendritic patches, the STED images are beautiful, but often too small. It is good to have a large overview of a substantial segment of dendrite, but zoom boxes would help distinguishing the small features and local organization of actin patches. A good

example of such zoomed regions is Fig. 4B - it would be good to generalize this to several other Figures describing the organization and partners of actin patches. For Figure 4B, the Results text says "LAMP1 puncta were distributed all over the dendrite" but these puncta are very difficult to see on the low-mag overlay image - perhaps panels with separated channels like on other Figures would help. Lastly, the 3D reconstruction from Fig. EV2B is surprising: in the view presented, all the synapses that appear over the shaft are "shaft synapses" (red). In a slice where environment is isotropic (no substrate effect), I would expect "spine synapses" (green) to be randomly oriented, and some of them to go upward and appear over the shaft on the view, rather than appearing all on either side, distant from the shaft. Could the authors present an additional orthogonal view of the same dendritic segment to clarify this?

Response: Following the reviewers suggestion we generalized the overview images with zoom boxes to Figure 2 and Figure 3 and added individual channels for LAMP1 and two channel overlays (LAMP1 with microtubules and LAMP1 with F-actin) in figure 4B. Regarding Figure EV2B: it is indeed the case that in organotypic slices (2-3 weeks in culture), synaptic spines tend to "fall" to the side and be present mainly orthogonally to the surface. We have included a 3D rotational video of the dendritic segment to showcase this (Movie 1).

3- Patches characterization - actin perturbations and cortactin presence

The Figure 3 and EV3 are showing the effect of actin-perturbing drugs on dendritic actin patches. The doses and treatment duration for each drug should be stated each time in the main Text/Figure/Legends (one or the other), rather than just as a table in the supplementary material. The effects are qualitatively reported as a more or less "reduction of F-actin at patches" - would it be possible to quantify this to make it more objective? Similarly, the colocalization of cortactin with actin patches should be quantified: what % of actin patches are cortactin-enriched? In line with point 1, is there a difference between spine base-associated and shaft synapse-associated patches for the effect of actin-perturbing drugs, and for cortactin enrichment (notably in relation to the cortactin enrichment at the base of spines described by Schätzle et al. *Curr Biol* 2018)?

Response: These are very valid points. We have included information on the drug concentration in the main text, the figures and corresponding legends. We have also repeated the pharmacology experiments 3 more times and quantified the change of phalloidin staining, size and counts for spine- and shaft-associated patches in the different drug treatment conditions using unprocessed STED data. Interestingly, we found that spine-base associated patches are more sensitive to the treatments, possibly due to a smaller size and missing association with the PSD. In the new experiments we also included cortactin staining and found that there is a correlation between the presence of cortactin and the patch size. We have fully updated Figure 3 and included the measurements of additional patch parameters in the Figure EV2C, D. As it can be seen from the images shown in Figure 2A, cortactin shows some enrichment in dendritic spines and patches, but it is also present in the shaft as diffusely distributed puncta. Cortactin is a very dynamic protein (Hering & Sheng, 2003; Mikhaylova et al, 2018) that re-distributes rapidly from the spine into the dendritic shaft upon synaptic activation. Our experiments were performed under basal conditions,

which means there will be spontaneous synaptic activity in these cultures. We have measured the intensity of cortactin staining upon perturbing F-actin and found that the correlation between the F-actin and cortactin signal intensity is no longer there after SMIFH2 and CK666 treatment. But we also noticed that there are very few spine-associated actin patches left after the LatA treatment, which makes it difficult to judge a correlation. We include these data as response to this reviewer but would like to exclude them from the manuscript.

4- Transport experiments – Generalization

Would it be possible to extend the results about the effect of actin patches on dendritic transport to other organelles or dendritic cargoes, beyond lysosomes and synthetic mobile peroxisomes, in order to support a general role? What happens for mitochondria?

Response: We believe that the general trafficking concepts described here can be extended to other secretory cargoes. However, each type of cargo will come with its unique characteristics (size, association with motor proteins, etc.) and investigating these for a whole new type of cargo would have gone beyond the scope of this manuscript. Nevertheless, as the reviewer suggested we looked into mitochondria trafficking. Mitochondria are highly mobile during neuronal development (van Spronson et al, 2014). We observed that mitochondria in mature neurons, as used throughout this study (DIV16-17), are present mainly as elongated, largely immobile structures that do not undergo trafficking events comparable to lysosomes and other small vesicular cargo. This observation is in accordance with (Rangaraju et al., 2019). An example image and kymographs derived from time-lapse recording are shown below.

5- Transport experiments - Streamlining of results

• The functional part of showing the effect of actin patches on the dendritic transport of organelles is interesting. However, in order to be more convincing, the results should be presented in a more simple and uniform way, without diluting the eventual differences in a large number of measured parameters. For example, for lysosomes, no effect is found for latA on the average velocity (Fig. 5D and 5I), but the speed is different in actin-rich regions (Fig. 6A). For PEX3-KIF17, by contrast, there is no difference in speed in actin-rich regions (Fig. 6C), but latA has a significant effect on the average velocity (Fig. 6E). The authors nonetheless state that the latA experiment performed on PEX3-KIF17 "address this apparent controversy" and "suggests, similar to lysosomes" [that have no difference in velocity after latA treatment] "an interaction with the actin cytoskeleton". Also, the PEX3-KIF17 data on the effect of LatA on velocity comes from a single experiment - replicates should be obtained.

Response: The Reviewer raises very valuable points. We performed additional experiments for lysosomes with DN myosins, myosin V and VI inhibitors and have included further independent experiments on the PEX3-KIF17 trafficking with Latrunculin A. Data analysis has been streamlined (please see reply to the next query for detail) and the text passages in question are now rephrased (highlighted in red).

• Quantification presented should be performed and presented in the same way across experiments. In particular, the quantification of pauses is quite confusing because it changes depending on the experiment and Figure. It is very difficult to assess and interpret what happens as the data are never presented with the exact same set of visualizations. Could the author provide the same set of quantification and visualization for all experiments: % of total time pausing/stationary/retrograde/anterograde (shown in Fig. 5B, 5G, 7F, 7I, EV5B, EV5F but not for PEF-Kif17 in Fig.6); % of pausing events (as in Fig. 7F, 7I, EV5B, EV5F but not for ctrl/latA in Fig.5 or PEX3-KIF17 in Fig. 6); number of immobile/mobile lysosomes per 10 μm (as in Fig. 7G, 7I but not for ctrl/latA in Fig.5, PEX3-KIF17 in Fig. 6 or myosin inhibitors on Fig. EV5); cumulative frequency of pausing times (shown in Fig.6G for PEX3-KIF17 but not Fig.5 for ctrl/latA, shown in Fig. EV5C and EV5G for myosin inhibitors but not in Fig.7 for myosin dominant negatives). Finally, I don't think the unique and odd "normalized pausing time" shown for PEX3-KIF17 in Fig. 6F should be kept or generalized to all relevant Figures.

Response: We have made the data representation of different experiments / figures uniform. Now for all interdependent experiments (pharmacological treatments of lysosomes and PEX-KIF17) we selected the analysis parameters, such as % of total time pausing/stationary/retrograde/anterograde, summed pausing time, relative frequency of pausing times, directional net flux, anterograde and retrograde velocity and the run length (Figure 5, Figure EV5, Figure 6). For the dominant negative myosins where different transfection groups were compared, we included the number of mobile and stationary lysosomes, % of total time pausing/stationary/retrograde/anterograde and relative frequency of pausing times (Figure 7). Additional parameters are shown in Figure S3. This should simplify the message and make groups and conditions more easily comparable between each other. Understanding the pausing behavior of organelles is one of the major points of this manuscript. Therefore, we put big emphasis on this parameter. We generalized the measurement of "summed pausing time" (e.g. Figure 6G) to all other relevant figures. This gives an idea about how treatments affected overall pausing time compared to control (100 %). Together with "pausing time, relative frequency" (e.g. Figure 6H), this provides the reader with information about total pausing time within the experiment and how these pausing events are distributed (short vs. long). Additionally, we described the different parameters in more detail in the Appendix.

6- Transport experiments - Role of myosins

• The data showing association of myosins with lysosomes is obtained on lysosomes purified from rat brain. Is it possible to show myosin presence on lysosomes in the same system as the rest of the study, i.e. in cultured neurons with immunocytochemistry and STED?

Response: As the reviewer suggested, we tried to stain endogenous myosin V and visualize it with STED microscopy, however, myosin V is present virtually all over the dendrite because it exists in cargo-associated and in soluble, diffuse forms. Unfortunately, it was not possible to draw conclusions on co-localization based on these images, please see the example below.

hippocampal primary cultures, DIV17

However, we found that expressed myoV DN (cargo-binding tail) partially co-localizes with LAMP1 labeled lysosomes (see Figure 7D).

- The authors should show the effect of myosin perturbation by DN or drugs on the correlation between transport parameters and actin patches localization: do myosin perturbations remove the lysosomes transport modification at actin patches shown in Fig. 6A-B?

Response: The experiment suggested by this reviewer is really interesting and could provide strong direct evidence for our hypothesis, but it is very hard to implement. In order to label F-actin, LAMP1/PEX-KIF17 and to inhibit myosins by dominant negative constructs, adult neurons would need to be transfected with three constructs at the same time. This is very challenging because of cytotoxicity issues which could alter the trafficking parameters on their own. One alternative could be to label F-actin with the fluorogenic probe SiR-actin. However this jasplakinolide-based drug stabilizes actin and we removed this from the revised version of the manuscript following suggestions of Reviewer 3.

- The results from the DN and inhibitors are not very clear - I'm not sure if the inhibitor data supports the conclusion that myosin V is implicated, but not myosin VI as the effect shown on Fig. EV5 are consistently more significant for TIP than for myovin for most parameters.

Response: We now increased the number of independent experiments for myosin V and VI DN constructs as well as the inhibitors and streamlined the analysis to make the data more uniform and easier to follow. The original statement still stands, myosin V has a clear effect on the stopping behavior of lysosomes, whereas effect of myosin VI is very minor. We find that TIP does not change the summed pausing time of lysosomes, which is one of the key measures for time being spent in an immobile state. TIP treatment had a significant effect on the processivity of lysosomes, specifically it decreased instant velocity and the run length. These parameters are mediated by microtubule motors like kinesins and dynein, and at the moment we cannot explain how TIP is influencing these motors. Nonetheless we decided to keep this information for the expanded view figure because it might be useful to other researchers.

- About the interpretation of the PEX3-KIF17 data as a demonstration of "passive" effect of actin patches: peroxisomes are immobile, but is it known that they don't have associated myosin V/VI on their surface?

Response: We have included immunostainings and a western blot of a peroxisome-enriched fraction that show the absence both myosin V and myosin VI (Figure EV4A, B).

Minor points

- When discussing the recently described actin structures in dendrites, it would be relevant to cite Nithianandam & Chien JCB 2018, which described actin "blobs" along dendrites that precedes dendritic branching.

Response: We tried to focus our manuscript on the role of dendritic F-actin patches in adult rat hippocampal neurons. The suggested reference points to the paper about actin waves in in developing DA neurons of *Drosophila* larvae. This is very different system and different developmental stage of neurons. Whereas the findings are very interesting, they are not within the scope of this study.

- The work of Schätzle et al. (Curr Biol 2018) is incidentally cited in the Discussion , but it clearly shows the presence of actin accumulation at the base of dendritic spines, so could be cited when discussing recently described actin structures in the Introduction.

Response: We have included a mentioning of this work in the introduction.

- In the Introduction, when discussing molecular processes underlying plasticity and stability of synaptic contacts, the surface diffusion of membrane components (such as ionotropic receptors) could be added.

Response: We added diffusion of membrane protein as a process relevant for synaptic plasticity and the corresponding reference in to the Introduction (Penn et al., 2017).

- The transfection times are indicated throughout the text, but I could not find what was the time between the transfection and the experiments themselves. Please precise this throughout or once in the Methods section if constant for all experiments.

Response: We have added the information about expression time / time after transfection (which was 1 day for all experiments) in the methods section.

- How were LatA or other drugs added in the live-cell imaging of transport experiments? Was a perfusion system used or manual addition of concentrated drug?

Response: We have added this information in the methods section.

- Results p.5, last line: "complimentary" seems be switched for "complementary"

Response: Thank you for noticing. The mistake was corrected.

Referee #2:

While the observation of actin patches within dendrites isn't novel, the manuscript breaks new ground by demonstrating a selective association of the patches with excitatory synaptic sites and providing support for a model where mobile dendritic organelles pause at these sites. However, I have a number of concerns that would need to be addressed before I would recommend publication in EMBOJ.

Major concerns:

-Overall there was little attempt to quantify the imaging data in figures 1-4. For example, what is the distribution of sizes of the actin patches? Does the actin patch size correlate with the size of the associated PSD? Can you estimate how close the edges of actin patches are to microtubules? The authors should use an established metric to describe how well actin patch signal correlates with various synaptic marker proteins.

Response: We thank this reviewer for raising this important point. A very similar concern was raised also by the first Reviewer. We have now provided further analyses comparing the sizes of the actin patches, as well as presence or absence of the post-synaptic density protein homer1 between spine-base associated vs. shaft-synapse associated patches. We discovered some distinguishing parameters between spine- and shaft-associated F-actin loci. For instance, homer1-positive patches in the shaft tend to be larger than homer1-negative patches; shaft-associated patches more frequently are homer1 positive than the spine-base associated F-actin. Now these new data are included in the Figure 2F. Regarding measuring the distance between the edges of actin patches and the microtubules, it is difficult to provide quantitative values as the STED data were acquired in 2D but not 3D mode. The measured distances would be very imprecise and subjected to overinterpretation and might not reflect the true situation. That's why we have chosen to demonstrate several qualitative examples instead.

- The synaptotagmin feeding experiments in Fig. 1C needs a negative control (preventing neural activity with TTX or similar during antibody exposure) to ensure the signal is specific to synaptic vesicle turnover. Also, showing only the chromobody/synaptotagmin channels here as a 2-color merge would be a cleaner way to demonstrate co-localization of functional terminals with postsynaptic actin since a significant fraction of the phalloidin signal likely comes from axons and presynaptic terminals. A separate set of images demonstrating colocalization between phalloidin and the chromobody could validate the chromobody labeling.

Response: Following the reviewer's suggestion we performed synaptotagmin antibody uptake experiments under basal conditions, and additionally with TTX as a negative control. We have replaced the old figure panel with new ones (Figure 2H), included an individual channel for

synaptotagmin as well as quantification of synaptotagmin labeling under basal and TTX treatment conditions.

Regarding the second point about validation of a dendritic origin of the actin signal, we now included two additional experiments. The first is 2 color STED showing that the expressed chromobody probe and phalloidin label the same patches in dendrites and spines (Figure 1B). The second is tagBFP-transfected neurons stained against F-actin, homer1 and gephyrin (3 color STED), showing that the patches localized inside of the dendrite filled with tagBFP (Figure 2C).

- Figure 2A relies on phalloidin staining to demonstrate patches colocalize with homer, but not gephyrin. I would also like to see this with an expressed marker for postsynaptic actin (e.g. sparsely expressed chromobody or lifeact) as some of the phalloidin signal could arise from presynaptic terminals.

Response: Please see reply above. We included two additional panels (Figure 1B and Figure 2C) to demonstrate dendritic and post-synaptic origin of the actin patches.

- The authors report widely ranging values and metrics for organelle mobility. I couldn't find any information about how these were calculated. These include "average speed" (e.g. Fig. 6B), "instant velocity" (e.g. Fig. 5I), and "average instant velocity" (e.g. Fig. 5D). How were these calculated and why do they give widely varying values (~0.1 micrometer/sec, average speed Fig. 6D vs ~1.2-1.5 micrometer/sec for "average instant velocity", Fig. 6E).

Response: Thank you for pointing this out. 'instant velocity' and 'average instant velocity' are the same, and are now referred to as 'instant velocity' throughout the manuscript.

In the revised version we included a better description of the kymograph analysis process, including definitions for 'instant velocity' and 'instant run length' (Methods section).

Only the analysis in Figure 6B, D uses parameter 'average speed'. For the analysis performed in Figure 6B, D, an elaborate description can be found in the method section of the Appendix. Supplementary Figure S2, provides graphical illustration of the workflow. Average speeds are low, since stationary events (0 $\mu\text{m/s}$) are included.

- It's hard to understand how organelle velocities were selectively measured within the actin patches since these structures are so small (near diffraction-limited). How many imaging frames did the organelles take to traverse these tiny compartments? How could the authors know the organelle was in contact with the actin patch since they were using diffraction-limited microscopy?

Response: Thank you for this question. The velocities of organelles are measured by the slope (displacement vs. time) of kymograph from the coordinates coming from lines manually traced in Fiji. The average speed was computed from all events inside and outside actin patches (defined in the kymograph analyses as 2D (x,t) rather than 3D (x,y,t)). Figure 6B, D illustrates that the presence of actin patches correlates with a lower speed of lysosomes. Since a randomization yields no significance difference between average speeds inside and outside actin patches, it's likely that actin patches induce pausing of organelles.

In this quantification actin patches are defined as ‘regions’. Indeed, due to the resolution limit, we cannot conclude that pausing is an effect of direct interaction of lysosome and F-actin. The information is extracted from the kymograph, where actin patch ‘regions’ are defined as x, t, while information about dendrite width (y) is largely lost. It is true that we might under-estimate the number of lysosome pausing events at actin patches, since lysosomes that seemingly pass by an actin patch without stopping might have simply not encountered that patch in 3D. In this case, the effect of the presence of actin patches would be even stronger. With the STED data (Figure 4), LatA treatments (Figure 5) and myosin inhibitor experiments (Figure 7), we are relative sure that the pausing events can result from direct interactions. Capturing these pausing events in 3D, with sufficient spatial and time resolution in one experiment would require a state of the art lattice light sheet – SIM system which recently was invented by the Betzig lab but is not available at our imaging facilities.

In general, the time (number of frames) that a lysosome/organelle takes to pass actin patches is very variable, which is also reflected in a *smooth curve* in the ‘Pausing time/ relative frequency (e.g. Figure 5K.)’.

• Treating the neurons with brefeldin A for 10h to conclude endosomes are not involved in actin patch generation seems like an indirect, and possibly misleading, experiment. I recommend omitting it.

Response: We included this experiment because of the previous report on axonal actin patches being nucleated from stationary endosomes (Ganguly et al., 2015). Similarly to this work, we used 10 h of brefeldin treatment and in contrast to the axonal patches, we did not find any noticeable effect on the dendritic F-actin loci, which suggests a different nature of these structures.

• While the actin patches appear stable over time, the authors should provide more quantitative dynamic measurements (using FRAP and/or single molecule tracking for example). Are actin patches near spines more or less dynamic? Are the PSD associated shaft patches more/less dynamic than neighboring non-PSD associated patches?

Response: This is a very good suggestion. To address this point we performed 3 types of experiments: **1)** repeated and quantified the effects of actin perturbing drugs (Figure 3A, B; Figure EV2C, D); **2)** did a FRAP experiment using chromobody-tagRFP; **3)** did a FRAP experiment using actin-RFP.

The first experiment revealed that upon treatment with Latrunculin A, the spine-base associated F-actin patches almost disappeared, whereas the shaft associated patches turned out to be more resistant, although their number was also significantly decreased (Figure 3A, B).

In the second experiment we used chromobody-tagRFP as an F-actin probe. Considering that chromobodies have their own F-actin association/dissociation kinetics, which were not described previously, we set out to characterize this probe in more detail and to test if it is suitable to address the above-mentioned questions. We began with HeLa cells because this cell line has very stable and prominent stress fibers with low F-actin turnover, which allows the analysis of the chromobody

turnover. Then we measured chromobody turnover in dendritic spines and dendritic patches of transfected neurons (see images and the analysis below). Analysis of the FRAP data indicated that the recovery rate of chromobody-tagRFP depended on the luminal volume: it was much faster in HeLa cells than in dendritic shafts, and showed the slowest recovery rate in segregated compartments such as dendritic spines. Therefore, the chromobody-tagRFP binding kinetics make it an unsuitable probe to study F-actin dynamics.

In the third experimental set we used low overexpression of RFP-actin. FRAP of over-expressed actin is a classical way to measure actin turnover in dendritic spines (Mikhaylova et al., 2018; Koskinen and Hotulainen 2014; Peris, Bisbal et al., 2018). RFP-actin was concentrated in dendritic spines and also labeled patches in the dendritic shaft. We performed FRAP analysis for dendritic spines and shaft-associated actin and found that, at both locations, actin showed very similar kinetics. These new data are now part of the Figure 2D, E. Unfortunately, we could not reliably measure FRAP of spine-base associated actin because in our imaging system the FRAP laser bleaches a spot about 1.5-2 μm in diameter. This also affects actin in the head of the associated dendritic spine and will make the interpretation of results more difficult. Since we did not find differences in FRAP between dendritic spines and shaft-associated patches, we would expect that spine-based associated F-actin would behave in a similar way. Concerning the influence of a PSD on the recovery rate of F-actin at the shaft-synapse-associated patches; there was very little variability in recovery rates of RFP-actin between all measured spots. According to our analysis about 65% of dendritic patches are PSD positive, thus many of the targeted regions contain PSDs. That's why we expect that the presence of a PSD will not have a big influence on the regulation of stable and dynamic actin pools, at least under basal synaptic activity conditions.

• In Fig. EV3B, how do you know SiR-actin labels only postsynaptic actin?

Response: This is a valid point. Also, considering the criticism of Reviewer 3 about the stabilizing effect of SiR-actin on the actin cytoskeleton, we removed these data from the manuscript and repeated the experiment using an actin-chromobody. Chromobody-TagRFP was co-transfected with eGFP as a dendritic volume marker. These data are now shown in Figure EV3A, B.

Minor issues:

• There is a callout to Fig. "S2" on pg. 8; there is no Fig. S2.

Response: We corrected this mistake.

• Scale bars are missing in some imaging panels (Fig 1B far right panels showing blown up images)

Response: We added the missing scale bars.

• On pg. 7 there is a reference to "axonal hotspot" with no context give for what this means

Response: We included a clarification in the text.

• It is not immediately apparent why randomizing the location of the actin patches in Fig. 6B (lower

panel) would have an "opposing effect" on organelle velocity since these patches presumably occupy a relatively small fraction of the total dendritic volume.

Response: We are sorry for this mistake, the randomization is actually non-significant (as also visible from the graph) rather than significant (**).

• No reference was included for the actin "chromobody" (Fig. 1C). I assume this is the commercially available reagent from Chromotek? References and details need to be included.

Response: Yes, this is the commercially available probe from Chromotek. We added this information to the Materials and Methods section.

Referee #3:

In this manuscript the authors propose interesting model where actin filaments act as molecular traps to impede the movement of endosomal vesicles (and perhaps help deliver these cargoes to locations within the dendrite). Though the question is an interesting one, there are significant conceptual and technical concerns that strongly limit enthusiasm for this model. Specific points are noted below.

1. A major issue with this manuscript is that it is not clear what the authors mean when they talk about "actin patches". Dendritic shafts have significant accumulations of actin, including enrichment in postsynaptic densities. The authors only provide a qualitative description without any serious attempt at clearly defining the very structures that they are describing. For instance in figure 1, there are numerous dots and patches of actin in the dendritic shaft, and the arrows only point to a few of them. Some of them look sub-plasmalemmal (and linear), while others are at the base of the spine and neck and look somewhat pyramidal in shape. Many other dots, spots and speckles are seen throughout the dendrite. In figure 1B it seems that the authors are pointing to postsynaptic densities, which are known to be enriched and actin. Indeed, the actin densities shown in the supplementary figure are clearly postsynaptic densities. How are these densities different from what the authors are calling "actin patches"or are they the same? No quantitative statements are made that strongly limits the impact of this work.

Response: We fully agree with this criticism also pointed out by Reviewers 1 and 2. In the revised version of the manuscript we now provide a detailed analysis of patch size, localization, association with the PSD and sensitivity to different pharmacological treatments affecting the F-actin pool. We discovered some distinguishing parameters between spine-base and shaft-associated F-actin loci. For instance, homer1-positive actin patches in the shaft tend to be larger than homer1-negative patches; shaft-associated patches are more frequently homer-1 positive than spine-base associated ones. The new data are included in the Figures 2, 3 and Figure EV2.

2. The older cultures are complicated, with numerous axons and dendrites intertwining - all of which have actin in them. How can the authors be sure that what they're looking at is definitely within the shaft of the dendrite? Though one realizes that this is a difficult problem to address, the onus is still on the authors to resolve this as clearly as possible.

Response: We agree with this point. We tried to address this issue more clearly by providing additional experiments using cell fills to outline the dimensions of the dendrites, and an expressed

actin marker, actin-chromobody-eGFP, where we clearly can say that the F-actin signal comes from within the dendritic compartment. These data are included into Figure 1B and Figure 2C. In addition, all STED data used for the analysis of patch size and localization also contained the dendritic microtubule marker MAP2, which allows judging whether a signal is inside or outside the dendritic volume.

3. The synaptotagmin uptake shown in figure 1C is not convincing.

Response: This point was also raised by Reviewer 2. In the revised version we performed the synaptotagmin antibody uptake experiments under basal conditions and upon silencing of synaptic activity using TTX. TTX application led to a reduced synaptotagmin uptake and staining intensity. We have replaced the old panel with new ones (Figure 2H), included an individual channel for synaptotagmin as well as quantification of synaptotagmin labeling at the basal and TTX treatment conditions.

4. For the authors model to be true, it is very important that the actin patches are not dynamic, however this is not been shown clearly. The authors use Sir-actin, which is a Taxol based probe that stabilizes actin, and it is not enough to conclude that the patches are stable. In fact the kymographs of actin chromobody imaging shown in figure 6 suggest that these patches are dynamic. The authors should have used a more physiologic marker like Lifeact to make this point. The polymerization of the actin structures upon latA (figure 3) also strongly indicate that these accumulations are dynamic.

Response: Following the Reviewer's recommendation, we repeated time-lapse imaging experiments in neurons transfected with chromobody-tagRFP to label F-actin and eGFP as volume marker. Similarly to SiR-actin we observed stable localization of patches at least over one hour of imaging period. These data now replace the initial SiR-actin experiment (Figure EV3A-B). We agree that describing the actin patches as "stable" was imprecise; in fact we meant to express that they are stably localized, while the F-actin turnover within the patch may well be very dynamic. We included additional experiments and quantifications (pharmacology and FRAP) and made this more clear in the revised manuscript (Figure 3). Regarding the actin dynamics, we found dendritic actin patches to be very similar to dendritic spines (Figure 3D, E). Considering a recent study indicating a stabilizing effect of Lifeact on F-actin (Flores et al, 2019), which confirms our personal experience with this probe, we decided not to include it in our study.

5. In figure 2E, there is no quantitative analysis.

Response: The reviewer raises an important point. However, we would like to note that we are actively investigating the molecular composition of excitatory shaft synapses as part of another larger study. As such, we are unable to directly address this comment here except for qualitative report that the shaft synapses also contain AMPA and NMDA receptors.

6. The quality of data in figure 4 is not very good, and it is difficult to agree with the author's conclusions (that are again qualitative).

Response: We disagree with this statement. In this figure we have depicted several different examples to demonstrate a spatial relationship between F-actin, microtubules and lysosomes in dendrites using 3 color STED microscopy, which is at the cutting edge of what is possible to achieve with super-resolution imaging techniques. Of note, the quality of this figure is acknowledged by Reviewer 1. The purpose of this figure is to show that microtubules do not invade actin patches and that lysosomes can be found in association with the F-actin mesh as well as with microtubules. As the story develops we demonstrate the contribution of both cytoskeletal structures in lysosomal trafficking using live imaging and pharmacological approaches followed by in depth analysis of the trafficking parameters. However, it is true that it indeed might have been difficult to appreciate small details in the overlay image in Figure 4B. We now separated the channels and included additional panels with F-actin + LAMP1 and tubulin + LAMP1.

7. Data in figure 5 is interesting, showing that the transport frequency of lysosomes and dendrites is increased upon treatment with lat A, and the anterograde bias is lost. However, it is unclear if these phenotypes have anything to do with actin patches. Why would the patches be involved in abrogating the anterograde bias of lysosomes? What is the effect of actin depletion on other dendritic cargoes? (the pex experiments are not useful). The concentration, time of lat A usage (and efficacy of actin-depletion in these older neurons) is also not clear.

Response: The loss of the anterograde bias is indeed interesting. It might have to do with the fact that most mature lysosomes originate in the soma and need to be transported anterogradely in order to reach distal dendrites. Perhaps actin patches serve as barriers to ensure proper distribution of lysosomes in dendrites. The observed increase in retrograde trafficking upon LatA treatment could mean that there might be differences in the dendritic retention of lysosomes driven by different motor proteins. For instance, we could show that KIF17-transported cargo is stalled at dendritic actin patches, but it could very well be that dynein or another kinesin motor can overcome such obstacles either due to their processivity characteristics or because they run on a different subset of microtubules.

In experiments requested by Reviewer 1 we extended our study to mitochondria (please see images provided for Reviewer 1). However, there are very few mobile mitochondria in dendrites of adult hippocampal neurons. This means that mitochondria are most likely not affected by presence of dendritic F-actin patches and multiple mechanisms could co-exist to regulate positioning of organelles in dendrites. These are very important questions that we would like to investigate in the future.

In addition, we have included information on LatA concentration and incubation time in the figures. As for the efficacy of F-actin depletion, in Figure 3 A-C and Figure EV2 C-D we show the significant reduction of F-actin structures upon LatA treatment.

8. Figure 6: it is not clear that the puncta seen by the actin chromobody really represents the actin patches (especially because the patches themselves are not well defined). The kymographs shown have much less LAMP GFP movements, compared to the other kymographs shown in the previous

figure. A critical unanswered question here is the frequency of vesicle movement. How many endosomes really pause at these patches? In many cases the slowing is very modest, and the statistical significance seems odd (Fig. 6B - below for example has **?).

Response: As the reviewers suggested, we improved the characterization of dendritic actin patches (please see comments above / Figure 2A-F) and provided new data showing the colocalization of an expressed actin-marker (chromobody) and phalloidin staining using confocal and STED microscopy (Figure 1A,B). From Figure 1B it is visible that patches are well detectable using confocal mode, but STED provides more precise measurements when it comes to the nanostructural characterization and size analysis.

The apparent visual differences between the kymographs in figures 5 and 6 stem from differences in time and size scale. We made sure to add the information about timing in all figure legends.

With regard to the quantification of lysosomes stalling at actin patches, we fully agree that the best readout would be to quantify the different trafficking parameters (pausing events, anterograde / retrograde transport and velocity) per each individual lysosome. Unfortunately, looking at the time-lapse movies and kymographs of lysosome trafficking, it is impossible to unequivocally follow and count individual lysosomes, (neither manually nor using various automated tracking programs), as their tracks overlap continuously. We did our best to quantify and distinguish lysosome trafficking behavior despite this draw-back using various selected parameters (summed pausing time, relative frequency of pausing events/time, total time spent in a stationary state).

As for Figure 6B – the Reviewer is correct in pointing out this mistake: in the earlier version of this manuscript, there were two asterisks mistakenly placed on the lower graph of Figure 6B, indicating a significance which was not there (as was stated correctly in the main text and figure legend). We now corrected the mistake. We included detailed description of the analysis in the Appendix and Figure S2.

In summary, there is some very impressive imaging in this manuscript, and the authors model is also an interesting one. The slowing of endosomal cargoes, and elimination of the bias upon actin depletion is also interesting, but it is really unclear the effect is due to the actin patches (or even what the actin patches really are). It is also unclear if the actin densities are really stable. Other points are noted above. Collectively, these concerns limit enthusiasm for this manuscript.

Response: We thank this reviewer for critical view on our manuscript which helped us to improve the data quality and to discover new details. We hope that with the revised version we adequately address his/her concerns.

Additional references:

Flores LR, Keeling MC, Zhang X, Sliogeryte K & Gavara N (2019) Lifeact-GFP alters F-actin organization, cellular morphology and biophysical behaviour. *Sci. Rep.* **9**: 1–13.

Hering H & Sheng M (2003) Activity-dependent redistribution and essential role of cortactin in dendritic spine morphogenesis. *J. Neurosci.* **23**: 11759–11769.

Mikhaylova M, Bär J, van Bommel B, Schätzle P, YuanXiang P, Raman R, Hradsky J, Konietzny A, Loktionov EY, Reddy PP, Lopez-Rojas J, Spilker C, Kobler O, Raza SA, Stork O, Hoogenraad CC & Kreutz MR (2018) Caldendrin Directly Couples Postsynaptic Calcium Signals to Actin Remodeling in Dendritic Spines. *Neuron* 97: 1110-1125.e14.

Rangaraju V, Lauterbach M, Schuman EM (2019) Spatially Stable Mitochondrial Compartments Fuel Local Translation during Plasticity. *Cell* 176: 73-84.e15.

Koskinen M, Hotulainen P (2014) Measuring F-actin properties in dendritic spines. *Front Neuroanat* 8: 74.

Peris L, Bisbal M, Martinez-Hernandez J, Saoudi Y, Jonckheere J, Rolland M, Sebastien M, Brocard J, Denarier E, Bosc C, Guerin C, Gory-Fauré S, Deloulme JC, Lanté F, Arnal I, Buisson A, Goldberg Y, Blanchoin L, Delphin C, Andrieux A (2018) A key function for microtubule-associated-protein 6 in activity-dependent stabilisation of actin filaments in dendritic spines. *Nature communications* 9: 3775.

2nd Editorial Decision

22nd May 2019

Thanks for submitting your revised manuscript to The EMBO Journal. Your study has now been re-reviewed by the three referees and their comments are provided below.

While referee #3 is still not convinced by some of the findings reported, referees # 1 and 2 appreciate the introduced changes and support publication here. Referee #1 has a few more comments that I would ask you to respond to in a last revision. Regarding the issues raised by referee #3 if you wish to respond to these with text changes that is fine with me, but also OK not to do so.

When you submit the revised manuscript would you also please sort out the things below:

REFeree REPORTS:

Referee #1:

In this revised version of their manuscript, the authors have addressed most of my points. I am happy to recommend acceptance for publication in the EMBO Journal after a few minor clarifications and corrections have been included:
-p.9, "we found that LAMP1-eGFP vesicles became more mobile following actin depolymerization (Fig.5H-K)": actually Fig. 5K does not show this, the latA effect shows no effect on pause length (even a non-significant tendency toward longer pauses).
- p.14, "life imaging" should be "live imaging".

Referee #2:

The authors have substantially revised their manuscript and have addressed my specific concerns with the initial submission. I am happy to support publication of the revised manuscript in EMBO J.

Referee #3:

Unfortunately this reviewer is not convinced that the data shown support the claims of this manuscript. The "actin patches" remain poorly defined (and scattered randomly throughout the

dendritic shaft, despite what the authors claim), and its difficult to even be convinced that these are really different from the other actin assemblies in dendrites. Some of them are linear (attached to the rings?), others are just dots and clusters. Though lysosomes appear to pause less upon Lat A treatment, it is unclear if the treatment even disrupts the so called patches. Fig. 6 seems to imply that the velocity of lysosomes is lower when they are within the patches but its difficult to see how this relates to the pausing behavior (pausing behavior or dwell-time inside/outside patches is not shown). No other cargoes are shown. The global effect of Lat A in neurons is also a problem in these experiments. The authors have obviously worked hard to gather the data, but in the opinion of this reviewer the data shown do not support the conclusions and the authors should take a critical look at their data, rather than trying to fit a model. However, if the other reviewers are satisfied, then please feel free to ignore these comments.

2nd Revision - authors' response

27th May 2019

The authors performed the requested editorial changes.

3rd Editorial Decision

29th May 2019

Thanks for sending us your revised version. I have looked at everything and all looks good.

I am therefore very happy to accept the manuscript for publication here.

Corresponding Author Name: Marina Mikhaylova

Journal Submitted to: EMBO J

Manuscript Number: EMBOJ-2018-101183R